



# Estimating turbulent energy flux vertical profiles from uncrewed aircraft system measurements: Exemplary results for the MOSAiC campaign

Ulrike Egerer[1,2, now at 7], John J. Cassano[1,2,3], Matthew D. Shupe[1,5], Gijs de Boer[1,5], Dale Lawrence[4], Abhiram Doddi[4], Holger Siebert[6], Gina Jozef[1,2,3], Radiance Calmer[1,2], and Jonathan Hamilton[1,5]

[1]Cooperative Institute for Research in Environmental Sciences, University of Colorado, Boulder, CO, USA
[2]National Snow and Ice Data Center (NSIDC), University of Colorado Boulder, Boulder, CO, USA
[3]Department of Atmospheric and Oceanic Sciences, University of Colorado, Boulder, CO, USA
[4]Smead Aerospace Engineering Sciences, University of Colorado, Boulder, CO, USA
[5]NOAA Physical Sciences Laboratory, Boulder, CO, USA
[6]Leibniz Institute for Tropospheric Research (TROPOS), Leipzig, Germany
[7]National Renewable Energy Laboratory (NREL), Golden, CO, USA

**Correspondence:** Ulrike Egerer (ulrike.egerer@nrel.gov)

**Abstract.** This study analyzes turbulent energy fluxes in the Arctic atmospheric boundary layer (ABL) using measurements with a small Uncrewed Aircraft System (sUAS). Turbulent fluxes constitute a major part of the atmospheric energy budget and influence the surface heat balance by distributing energy vertically in the atmosphere. However, only few in-situ measurements exist of the vertical profile of turbulent fluxes in the Arctic ABL. The study presents a method to derive turbulent heat

fluxes from DataHawk2 sUAS turbulence measurements, based on the flux gradient method with a parameterization of the turbulent exchange coefficient. This parameterization is derived from high-resolution horizontal wind speed measurements in combination with formulations for the turbulent Prandtl number and anisotropy depending on stability. Measurements were taken during the MOSAiC expedition in the Arctic sea ice during the melt season of 2020. For three example cases from this campaign, vertical profiles of turbulence parameters and turbulent heat fluxes are presented and compared to balloon-borne,

radar and near-surface measurements. The combination of all measurements draws a consistent picture of ABL conditions and demonstrates the unique potential of the presented method for studying turbulent exchange processes in the vertical ABL profile with sUAS measurements.

## 1 Introduction

This work analyzes turbulent energy fluxes in the Arctic atmospheric boundary layer (ABL), based on measurements with a

small Uncrewed Aircraft System (sUAS). The Arctic ABL interacts with the underlying sea ice by modulating the surface energy budget. Turbulent processes, in particular turbulent energy fluxes, play a major role in the ABL development, because they describe how energy is distributed vertically within the ABL. Turbulent fluxes of sensible and latent heat and momentum are intertwined with cloud formation, the movement of sea ice, and other key interactions between the atmosphere and surface.



The Arctic ABL is typically stratified in terms of temperature, humidity, aerosol concentration, etc., and knowing the vertical
profile of turbulent fluxes sheds light on how these different layers interact.

In the central Arctic ABL, very few in-situ vertical profile observations of turbulence parameters exist. Vertical profile measurements of turbulent energy fluxes are even less common because they require high-resolution and accurate measurements of the vertical wind velocity and other atmospheric state parameters. Turbulent energy fluxes have been estimated from sophisticated aircraft-based measurements of the three-dimensional wind vector (e.g., Tjernström, 1993), but only for limited time
periods due to expensive aircraft operation and extensive organizational efforts. sUAS are more convenient to operate and are increasingly used, especially for turbulence observations (e.g., Kral et al., 2020; Lampert et al., 2020; de Boer et al., 2018). The low flight speed has less impact on the measured turbulence parameters and the high vertical resolution is beneficial for studying thin layers of turbulence (Balsley et al., 2018). Fixed-wing aircraft make use of spiral ascents or slant profiles. Further, sUAS can fly at very low altitudes, which is advantageous for studying the shallow Arctic ABL (Jonassen et al., 2015) and
its interaction with the surface. sUAS-based high-resolution turbulence measurements are usually obtained with pitot-static or multi-hole pressure probes (van den Kroonenberg et al., 2008; Calmer et al., 2018; Kral et al., 2020) and turbulence parameters have been derived from those measurements (Balsley et al., 2018; Luce et al., 2019). If the three-dimensional wind vector is measured by the multi-hole probe, turbulent fluxes can be directly estimated (Rautenberg et al., 2019). However, sUAS observations often provide the one-dimensional horizontal wind velocity relative to the instrument, which requires further con-
siderations to estimate turbulent fluxes. For example, Knuth and Cassano (2014) apply an integral method to retrieve the fluxes from mean quantities, Båserud et al. (2019) derive fluxes from several consecutive vertical mean profiles, and Greene et al. (2022) make use of mean gradient-based similarity functions in the stable Arctic ABL.

The year-long field campaign MOSAiC (Multidisciplinary drifting Observatory for the Study of Arctic Climate;  Shupe et al., 2022) is a unique opportunity for detailed observation of Arctic ABL conditions. During MOSAiC, the DataHawk2
(DH2) sUAS (Lawrence and Balsley, 2013; Hamilton et al., 2022) was operated to measure the horizontal wind velocity and temperature with a high temporal resolution. Turbulence parameters have been derived from DH2 measurements (Luce et al., 2019; Balsley et al., 2018), but the derivation of turbulent fluxes has not been studied in detail, as the vertical wind velocity for the direct estimate of turbulent fluxes based on the eddy covariance method is not measured directly. The present paper proposes the flux gradient method as an alternative method to estimate the turbulent fluxes from DH2 measurements. With a
parameterization of the turbulent exchange coefficient based on turbulence estimates, the vertical profile of turbulent fluxes of latent and sensible heat is reconstructed. The parameterization must be suitable for the Arctic ABL conditions. During MOSAiC, alongside the DH2 the tethered-balloon system BELUGA (Balloon-born moduLar Utility for profilinG the lower Atmosphere) was operated and provides direct turbulent flux estimates by measuring the earth-referenced, three-dimensional, high-resolution wind vector (Egerer et al., 2019) as a reference. The sUAS-based flux profiles are further evaluated together
with surface-based measurements of turbulent fluxes as a continuous and well-characterized perspective. The present paper is structured as follows: Section 2 introduces the field campaign and measurement platforms, the applied method for turbulent flux estimation, and elaborates how DH2 measurements are applied with this method. Section 3 presents vertical profiles of turbulent parameters and fluxes for three example cases for stable stratification, a decoupled mixed-phase cloud layer, and a



cloud- and wind-shear driven ABL and compares DH2 observations to BELUGA and mast measurements. The limitations and
future potential of the applied flux method are discussed in Sect. 4.

## 2 Methods

### 2.1 Measurement platforms and campaign

#### 2.1.1 MOSAiC campaign

During the year-long MOSAiC expedition, the German icebreaker *Polarstern* (Knust, 2017) was frozen in the sea ice and
drifted across the Arctic Ocean between September 2019 and September 2020 (Shupe et al., 2022). Extensive measurements
were taken to explore the Arctic climate system, including the ocean, sea ice, and atmosphere. The drift expedition is divided
into five legs, covering the annual cycle, and includes measurements on the ship and on/below/above the ice surrounding the
ship.

The DH2 sUAS was flown from March to August 2020, during legs 3 and 4 of the MOSAiC field campaign, over the Arctic
Ocean ice pack (de Boer et al., 2022). The flights were conducted on the sea ice adjacent to the icebreaker *Polarstern* and the
sea ice cover changed from primarily snow-covered with some ridges and leads to bare ice with melt ponds over the 5-month
period. The tethered balloon system BELUGA was operated during leg 4 in July 2020 (Lonardi et al., 2022) from the ice floe
150 m to 700 m apart from the DH2. The instruments on the balloon recorded vertical profiles of thermodynamics, aerosol
particles, clouds, radiation, and turbulence properties. A summary of all flights is provided in de Boer et al. (2022) and Lonardi
et al. (2022), respectively. Additionally, a meteorological mast on the ice floe recorded meteorology and wind conditions at
different heights close to the surface (Cox et al., 2021). The temporal development of clouds can be evaluated by means of
cloud radar (Johnson et al., 2021) and ceilometer measurements made onboard *Polarstern* (Schmithüsen, 2021).

On four days in July 2020, the two airborne platforms were operated nearly simultaneously. The present work includes these
comparison days (characterized by mostly stable stratification) and a DH2 flight on 9 April 2020 in less stable conditions.
Flight profiles for the days studied are shown in Fig. 1 together with the cloud boundaries. Whereas the DH2 measurements
stop at cloud base in most cases, BELUGA adds in-cloud measurements above the DH2 profiles.

#### 2.1.2 DH2 small uncrewed aircraft system

sUAS fill a niche in atmospheric observing, offering perspectives that are challenging to obtain with other in situ sensing meth-
ods. This includes an ability to fly in a variety of atmospheric conditions, some of which (e.g. fog, low aerosol concentrations,
no clouds/precipitation) can challenge traditional remote sensing techniques. sUAS can provide observations at altitudes from
single meters above the surface all the way up through the upper troposphere. Additionally, such platforms can provide high
temporal and spatial resolution, and resample a given layer of interest repeatedly. Their ability to sample horizontally also
offers unique perspectives on spatial heterogeneity. While these platforms offer numerous advantages and can capture unique





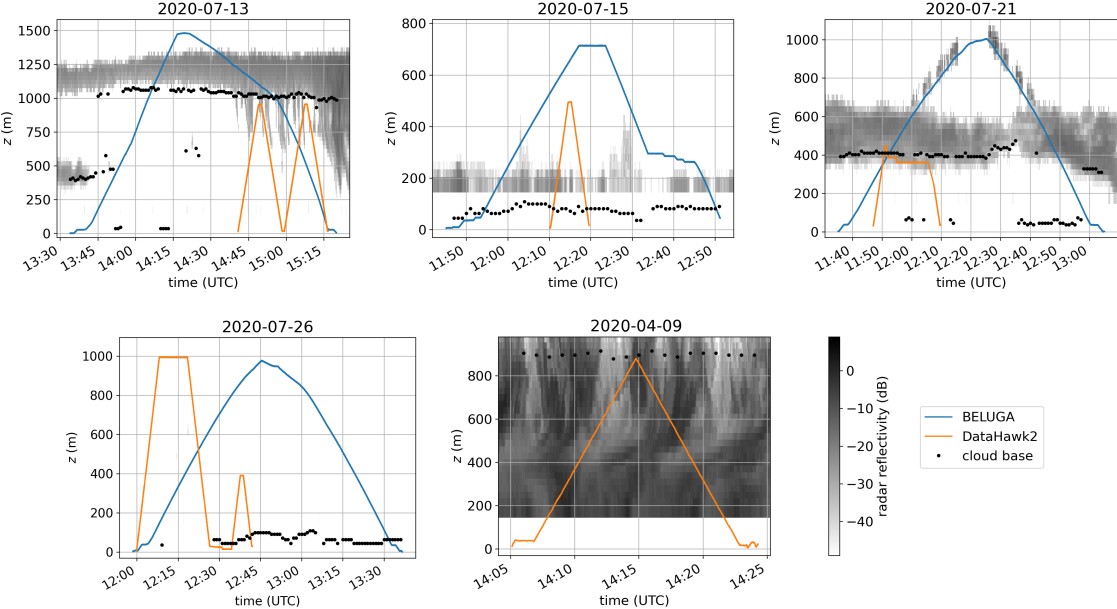

**Figure 1.** Time-height profiles of DH2 and BELUGA flights for days presented in this paper. Days in July 2020 include the cases when both platforms were operated simultaneously. Cloud base is from the *Polarstern* ceilometer (Schmithüsen, 2021) The gray shading shows radar reflectivity (Johnson et al., 2021). Note that the BELUGA balloon was observed by the radar during the case on 21 July 2020.

information, there are also constraints to their operation related to weather (winds, visibility, icing conditions) and specific
regulations.

The DH2 is instrumented to collect detailed information on the thermodynamic structure of the atmosphere, while simultaneously collecting data on winds and atmospheric turbulence. To observe the thermodynamic state, the system was equipped with a Vaisala RSS-421 pressure, temperature, and humidity sensor suite, with sensors extending into the streamflow passing over the aircraft. The platinum resistive temperature sensor on the RSS-421 offers 0.01 °C resolution and a repeatability of
0.1°C with a 0.5 s $1/e$-response time at typical airspeeds. This sensor also includes a capacitive silicon pressure sensor offering 0.01 hPa resolution and a 0.4 hPa repeatability, and a thin-film capacitive relative humidity (RH) sensor that offers a resolution of 0.1 % RH, and a repeatability of 2 % RH. The response rate of the RH sensor is temperature-dependent, and ranges from approximately 0.3 s (at 20 °C) to 10 s (at -40 °C). In addition to the Vaisala sensor, the DH2 is equipped with a custom finewire sensor. This consists of 5 µm diameter platinum sensor wires, with one operated as a coldwire thermometer and the
other heated to serve as a hotwire anemometer. The finewire array also includes a Sensiron SHT-85 temperature and humidity sensor. A pitot-static probe serves as a reference for the hotwire anemometer. Finally, the DH2 is equipped with up- and downward-looking IR thermometers that offer qualitative information on the surface state under the aircraft and the presence of clouds above the aircraft, as well as a custom-designed sensor suite to measure aircraft attitude, position, and velocity. More details on the DH2 platform can be found in Hamilton et al. (2022).



During MOSAiC, this platform was operated semi-routinely from the expedition's central observatory, near the frozen-in icebreaker *Polarstern*. The aircraft conducted frequent profiles to one kilometer above the ice surface, operating in a spiral ascent/descent pattern while approximately maintaining a single geodetic location above the slowly drifting ice pack. For this project, operations were limited to time periods when average winds were below $10\,\mathrm{m\,s^{-1}}$ and visibility was sufficient to keep track of the aircraft position in flight. This prevented the aircraft from flying in significant precipitation and/or through clouds.

The DH2 was operated in very cold temperatures (down to -37 °C) and over broken sea ice and melt-pond-covered surface. Despite weather challenges, the DH2 conducted a total of 82 flights during MOSAiC, compiling 42.9 flight hours of data over the central Arctic Ocean. Additional details on the MOSAiC DH2 deployment, including photographs of the aircraft, can be found in de Boer et al. (2022), and the data from these deployments are publicly available through the NSF Arctic Data Center (Jozef et al., 2022b, 2021).

The pitot-static probe and the finewire sensors have been used in previous studies (e.g., Kantha et al., 2017; Balsley et al., 2018; Luce et al., 2019) to derive turbulent parameters such as the temperature structure parameter, eddy dissipation rate and Ozmidov scale. In the present study, we use the pitot-static probe and the hotwire probe for deriving turbulence parameters. The pitot-static probe, connected to a differential pressure sensor, is calibrated post-flight as described by Doddi et al. (2022). The procedure for calculating horizontal winds from the pitot airspeed is also described in this study. The hotwire airspeed

cannot be calibrated directly to the pitot airspeed because the zero-voltage of the measurement circuit is adjusted in-flight to accommodate the measurement range. Therefore, hotwire airspeed fluctuations are calibrated to pitot airspeed fluctuations in the spectral space for defined time intervals of 2 s. Figure 2 gives an example of a 2 s spectrum for pitot and hotwire fluctuations. The pitot spectrum typically shows spikes and artifacts at frequencies above around 10 Hz due to motor noise and vibrations, which has been observed as well in previous studies (e.g., Luce et al., 2018). The spectral peaks are more pronounced on

ascents. The hotwire spectra are much less distorted by motor vibrations.

The hotwire spectrum is calibrated to the pitot spectrum with a calibration constant for each individual spectrum. Spectral peaks for the pitot are excluded at frequencies where the standard deviation from a $f^{-5/3}$ slope exceeds a defined threshold. For MOSAiC, this calibration method provides a hotwire calibration coefficient that varies more than anticipated (within time periods where it is expected to be constant) and significantly influences the calibrated hotwire time series. Therefore, the

calibration method can only be applied to retrieve hotwire spectra, but not to retrieve calibrated hotwire airspeed fluctuations. Turbulence parameters in this paper will be mainly derived from the pitot airspeed, as discussed in Sect. 2.3.

### 2.1.3 Tethered balloon system

The tethered balloon system BELUGA consists of a helium-filled tethered balloon attached to a winch via a 2000 m, 3-mm-thick Dyneema cable. The nominal volume of the balloon is 90 m³, yielding a free buoyancy at the ground level of about

40 kg, of which 15 kg is available for scientific payload. The balloon can be safely operated under light icing conditions at wind speeds up to $15\,\mathrm{m\,s^{-1}}$ at higher altitudes and about $5\,\mathrm{m\,s^{-1}}$ near the ground. In general, ascent and descent rates of approximately $1\,\mathrm{m\,s^{-1}}$ allow a complete vertical profile to 1 km altitude (ascent and descent) in about 30 minutes. In addition to a small probe for measuring standard meteorological parameters with a ground connection realized by a radiosonde module,



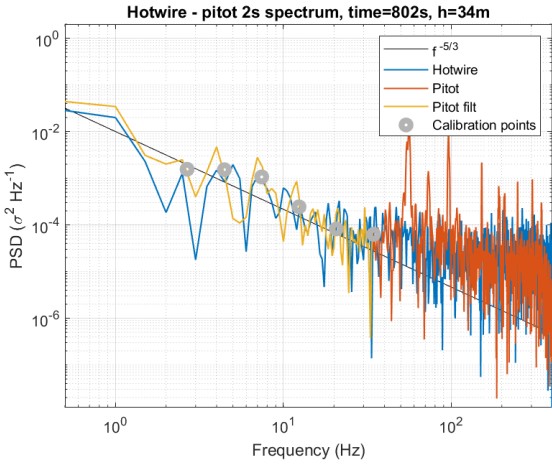

**Figure 2.** Comparison of hotwire and pitot spectra for one 2 s segment on 13 July 2020. The hotwire fluctuations are calibrated to the pitot fluctuations in the frequency range marked by the gray dots ("pitot filt").

the main instrument of this data analysis is a turbulence probe based on an ultrasonic anemometer in combination with an
attitude reference system (Egerer et al., 2019).

The temporal resolution of the three-dimensional wind vector is 50 Hz, which corresponds to a typical spatial resolution of
10 cm at 5 m s$^{-1}$ mean wind speed. Using attitude angles and inertial velocities, the observed wind vector is transferred to
an earth-fixed reference system. An advantage of the ultrasonic system is the additional measurements of virtual temperature,
which allows direct measurement of turbulent heat and momentum fluxes. The application and the limitations for turbulent flux
estimates with BELUGA are discussed in Egerer et al. (2019, 2021).

### 2.1.4 Additional measurements

Vertical profile measurements at lower altitudes can be compared to near-surface meteorology and flux measurements at nominal heights of 2 m, 6 m, 10 m and temporary 23 m made from towers mounted on the sea ice within 300–400 m distance
of *Polarstern* (Shupe et al., 2022). These surface-based measurements provide a reference for flux magnitudes using a well-
accepted ground-based eddy-correlation approach. Measurements of interest for this study were made by temperature and
relative humidity probes and sonic anemometers mounted at all heights. The sonic anemometers operated at 20 Hz and were
resampled to 10 Hz for analysis. Additionally, surface pressure was measured at the 2m height and 10 Hz measurements of
water vapor concentration were made at 2 m (May 2020 and earlier) or 6m heights (June 2020 and after). Collectively these
measurements were used to derive surface sensible, latent, and momentum fluxes via the eddy correlation technique at nominal
10-min intervals using 13.6 min segments of data (Cox et al., 2021). The 23m height was only available during the period
up until May. For the period starting in June, the meteorological tower was installed approximately 100m from the BELUGA
launch site during June-July.





Additionally, the Ka-band ARM Zenith Radar (KAZR) was operated by the US Department of Energy (DOE) Atmospheric Radiation Measurement (ARM) program onboard *Polarstern* (Johnson et al., 2021). It provided continuous vertical mea-
surements of the radar reflectivity, mean Doppler velocity, and spectral width, which collectively provide information on the vertical distribution of clouds, the type of clouds, and the presence of turbulent mixing in the atmosphere. The radar and mast measurements serve as a reference and for putting the DH2 measurements into context.

## 2.2  Flux gradient method and turbulent exchange coefficient

The flux gradient method (Stull, 1988), as a first-order local turbulence closure scheme, relates local gradients and respective
turbulent fluxes. Using this method, turbulent fluxes (e.g., the turbulent sensible heat flux $\sim \overline{w'\theta'}$) can be approximated from vertical profiles of mean parameters (marked with an overline):

$$\overline{w'\theta'} = -K_{\mathrm{H}} \cdot \frac{\partial \overline{\theta}}{\partial z}, \tag{1}$$

using the mean vertical potential temperature gradient $\partial \overline{\theta}/\partial z$. The turbulence exchange coefficient for heat $K_{\mathrm{H}}$ must be parameterized as a function of the flow. Commonly, the parameterizations are formulated for the turbulent exchange coefficient
of momentum $K_{\mathrm{m}}$ (Holt and Raman, 1988), which relates to $K_{\mathrm{H}}$ via the turbulent Prandtl number $\mathrm{Pr}_t$:

$$\mathrm{Pr}_t = K_{\mathrm{m}}/K_{\mathrm{H}}. \tag{2}$$

Similarly, the turbulent latent heat flux is related to the mean profile of specific humidity $q$:

$$\overline{w'q'} = -K_Q \cdot \frac{\partial \overline{q}}{\partial z}, \tag{3}$$

with $K_{\mathrm{Q}} \approx K_{\mathrm{H}}$ (Dyer, 1967) and the mean humidity gradient $\partial \overline{q}/\partial z$. The flux gradient method is one of the simplest turbulence
parameterizations and is particularly suited for small eddies (Stull, 1988). The presence of larger-sized eddies and counter-gradient fluxes might cause the method to fail. However, a local closure scheme (where $K$ is a local estimate) might be best suited to describe a non-classical, complex ABL with e.g. multiple inversions.

A large number of parameterizations for $K_{\mathrm{m}}$ have been developed (e.g., Bhumralkar, 1976; Mahrt and Vickers, 2003; Cuxart et al., 2006) and are widely used for sub-grid turbulence in models. Some of the parameterizations are derived from
or validated against airborne measurements (e.g., Bélair et al., 1999; Aliabadi et al., 2016). The main method for the $K$ parameterization used in the present work is based on the work of Hanna (1968), who suggested parameterizing $K$ based on local turbulence parameters. The study, based on dimensional reasoning, hypothesizes that $K$ can be parameterized by the quantities determining the turbulent energy spectrum of vertical wind velocity: standard deviation $\sigma_w$, eddy dissipation rate $\varepsilon_w$ and the wavelength of the peak in the wind velocity energy spectrum. As these parameters inter-depend (Hinze, 1975), two
out of the three are sufficient to determine $K$. Applying this parameterization allows deducing turbulent fluxes from vertical profiles of the measured turbulence parameters $\sigma_w$ (or variance $\sigma_w^2$) and $\varepsilon_w$. $K_{\mathrm{m}}$ is related to $\sigma_w$ and $\varepsilon_w$ by:

$$K_{\mathrm{m}} = C \cdot \frac{\sigma_w^4}{\varepsilon_w}, \tag{4}$$





leading to

$$K_{\mathrm{H}} = C \cdot \frac{\sigma_w^4}{\varepsilon_w} / \mathrm{Pr}_t. \tag{5}$$

The turbulent Prandtl number $\mathrm{Pr}_t$ can be approximated (Sect. 2.3.4); $C$ is a constant with $C = 0.35$ for nearly-neutral conditions. In the original paper, the parameterization is validated by different observational data over land and sea from towers and aircraft in various stability regimes in the ABL up to 320 m height.

With the parameterized turbulent exchange coefficient, the turbulent fluxes of heat and moisture result as

$$H_{\mathrm{S}} = -\overline{\rho} \cdot c_{\mathrm{p}} \cdot K_{\mathrm{H}} \cdot \frac{\partial \overline{\theta}}{\partial z} \tag{6}$$

$$H_{\mathrm{L}} = -\overline{\rho} \cdot L_{\mathrm{v}} \cdot K_Q \cdot \frac{\partial \overline{q}}{\partial z} \tag{7}$$

where $\rho$ is the air density, $c_{\mathrm{p}}$ is the specific heat capacity of air, $L_{\mathrm{v}}$ is the latent heat of vaporization, and assuming $K_{\mathrm{Q}} \approx K_{\mathrm{H}}$. The method by Hanna (1968) has been used for a wide range of applications. It is often adopted for air pollution modeling (Tomasi et al., 2019; McNider and Pour-Biazar, 2020), and has served to calculate particle fluxes based on sUAS data (Platis et al., 2016). The method has even been extended to ABL conditions in a tropical cyclone (He et al., 2021) using tower
observations and hurricane conditions in the low-level troposphere (Zhang et al., 2010) using aircraft observations.

To apply the Hanna (1968) parameterization, some assumptions have to be made. First, $\mathrm{Pr}_t$ is a function of stability (Li, 2019): heat transport is suppressed under stable conditions through buoyancy effects. Different formulations exist for how $\mathrm{Pr}_t$ varies depending on different stability parameters and a relationship has to be selected based on available data and conditions (see discussion in Sect. 2.3.4). Second, some studies use revised values for the constant $C$ (e.g., $C$=0.41; Nieuwstadt, 1984;
Zhang et al., 2010). However, we stick to the original value $C = 0.35$ (Hanna, 1968; Busch and Panofsky, 1968). Last, it matters if the turbulence parameters in Eq. (4) are measured in the vertical direction or along the mean flow because of anisotropy of the flow. Ideally, in an isotropic flow, the statistical properties of turbulence parameters are independent of their direction: $\sigma_u^2 = \sigma_v^2 = \sigma_w^2$; for local isotropy in the inertial subrange the turbulent fluctuations in this region scale as $\sigma_v^2/\sigma_u^2 = \sigma_w^2/\sigma_u^2 = 4/3$ (Kolmogorov, 1941; Busch and Panofsky, 1968; Kaimal et al., 1972). Because atmospheric turbulence
at larger scales is predominantly anisotropic (e.g., Lovejoy et al., 2007; Biltoft, 2001), this has to be considered when measuring turbulence. The next section introduces the DH2 measurements, and how they can be applied with the flux gradient method and discusses the assumptions mentioned above.

## 2.3    Application of the flux gradient method to DH2 measurements

The DH2 provides measurements of the one-dimensional horizontal airspeed with a pitot-static probe and a hotwire anemome-
ter at a sampling frequency of $f = 800$ Hz. These measurements provide the basis to apply the flux gradient method with the $K$ parameterization by Hanna (1968) based on the turbulence parameters dissipation rate ($\varepsilon$) and wind velocity variance ($\sigma^2$). Fluctuations in the measured airspeed of the pitot and hotwire probe are assumed to correspond to fluctuations in the actual wind velocity.



### 2.3.1 Dissipation rate

The turbulent energy dissipation rate $\varepsilon$ is of central importance in describing turbulent flows. Muschinski et al. (2004) and
Siebert et al. (2006) discuss different methods for estimating local dissipation rates from airborne in-situ measurements. Most
commonly, $\varepsilon$ is derived from the energy spectrum and from structure functions.

The spectral method is based on the turbulent energy spectrum of vertical or longitudinal wind velocity fluctuations in a time
period $\tau$. In the inertial subrange, the energy spectrum has the universal form

$$E(k) = \alpha \cdot \varepsilon^{2/3} \cdot k^{-5/3} \tag{8}$$

(Kolmogorov, 1941) with the energy dissipation rate $\varepsilon$, wave number $k = \frac{2\pi \cdot f}{\overline{U}}$, mean horizontal airspeed $\overline{U}$ and the Kol-
mogorov constant $\alpha \approx 0.5$ for the longitudinal wind velocity (Högström et al., 2002; Yeung and Zhou, 1997). A fit to the
energy spectrum of a measured time series provides $\varepsilon$.

Structure functions are based on the velocity increment $u(t - t^*) - u(t)$ between two measurement points at times $t - t^*$
and $t$. By evaluating the structure function for a discrete time series of a flow velocity component, the dissipation rate can be
retrieved. Siebert et al. (2006) concluded that the second-order structure function provides the most robust results for estimating
local dissipation rates from observational data. We estimate local dissipation rates $\varepsilon_\tau$ from the second-order structure function

$$D^2(t^*) \equiv \overline{[u(t - t^*) - u(t)]^2}_\tau = C_2 \cdot \varepsilon_\tau^{2/3} \cdot \left(t^* \cdot \overline{U}_\tau\right)^{2/3} \tag{9}$$

in a time period $\tau$ with $C_2 \approx 2$ for the longitudinal flow component ($C_2 \approx 2.6$ for vertical velocity components). Averaged
parameters in Eq. (9) are indicated by an overline and time interval index $\tau$, $u(t)$ is the horizontal wind velocity at the time $t$,
and $t^*$ is a time lag.

For each of the DH2 and BELUGA platforms, a different established method is applied to derive dissipation rates. The
DH2 turbulence dissipation rates are derived from high-resolution pitot airspeed fluctuations following the turbulence spectral
analysis presented in previous publications (Frehlich et al., 2003; Luce et al., 2018, 2019; Doddi et al., 2022). The dissipation
rates are computed by fitting the measured one-dimensional airspeed frequency power spectra to the model universal turbulence
energy spectrum $E(k)$ in the inertial subrange characterized by a $k^{-5/3}$ slope (Kolmogorov, 1941). Taylor's frozen flow
hypothesis (Pope, 2000) is invoked to approximate the temporally measured DH2 one-dimensional airspeed power spectra as
wavenumber spectra thereby enabling model spectral fitting. First, the pitot-measured airspeed data are segmented into 2 s
interval time records. The time records are detrended and subsequently windowed by a variance-preserving Hanning window.
A fast-Fourier transform algorithm is implemented to compute energy spectra of the windowed time records and normalized to
obtain the one-dimensional airspeed Power Spectral Density (PSD). The pitot-measured airspeed is contaminated by aircraft-
motor-vibrations-induced periodic artifacts appearing as sharp peaks in spectra at high frequencies (see Sect. 2.1.2 and Fig. 2).
The average PSD in equally spaced logarithmic frequency bins is computed to reduce the spectral variance and aid in identifying
the motor-induced periodic artifacts. The bin-averaged PSD is weighted by $f^{5/3}$ and subject to a spectral fitting procedure.
Discounting the prominent periodic artifacts identified, the mean and standard deviation of the $f^{5/3}$-weighted spectra are
calculated in a preset frequency range (2–400 Hz). The turbulence dissipation rates including error bars are estimated from





the spectral fit mean and standard deviation using Eq. (5) in Luce et al. (2019) and Eq. (30) in Frehlich et al. (2003). Careful consideration of the assumptions involved in turbulence spectral analysis including the details of the DH2 pitot spectral analysis procedures are presented in Doddi (2021). This spectral method provides dissipation rates from pitot airspeed fluctuations. The

hotwire provides comparable dissipation rates to the pitot because the hotwire spectrum for each segment is fitted to the pitot spectrum, however the varying HW calibration coefficient influences the results. Figure 3 shows a vertical profile of $\varepsilon$ for a day where the HW calibration was reliable. The spectral method for pitot and hotwire (black and blue crosses) show a very similar vertical structure and magnitudes. At smaller scales of $\varepsilon$ values, the two methods deviate from each other probably because there are fewer spectral averages above the noise floor and less reliable inertial range fits.

For BELUGA, the dissipation rates are derived from the second-order structure function (Egerer et al., 2019, 2021; Lonardi et al., 2022). For defined time periods $\tau = 2$ s, the structure function on the left side of Eq. (9) is evaluated for time lags $t^*$ in an empirical time range: $0.002$ s $< t^* < 1$ s. Fitting this curve to the right side of the equation yields $\varepsilon_\tau$ for each time period. Exponents (that should theoretically equal 2/3) are accepted in a range of 0.3 to 0.9, otherwise no dissipation rate can be estimated. Since the sonic anemometer provides the three-dimensional wind vector, dissipation rates can be estimated

for the horizontal and vertical wind components. For BELUGA, the vertical wind component is used because of the higher measurement resolution. However, Hanna (1967) found it easier to determine $\varepsilon$ from the horizontal component rather than from the vertical and found the relation $\varepsilon_u = 1.6 \cdot \varepsilon_w$ near the ground with similar values higher up. The issue of anisotropy will be further discussed in Sect. 2.3.5.

For BELUGA and DH2, the established method is used because it is suited to the individual characteristics of the respective

measurements in terms of distinctive features in the spectra, measurement resolution, etc. To exclude errors resulting from different procedures, both methods are compared by applying the structure-function approach to the DH2 data. Because of the artifacts in the pitot fluctuation time series, hotwire fluctuations restored from the fitted spectra are used. This is possible for the 26 July flight, which had few variations of the hotwire calibration constant. The results in Fig. 3 suggest that both the structure function and spectral method provide similar $\varepsilon$ results for the DH2. The spectral method for pitot will be used as the

default method for the DH2. The results for BELUGA are added for comparison, showing a comparable vertical structure with a similar magnitude of $\varepsilon$. BELUGA dissipation rates are slightly higher than DH2 values at lower altitudes and lower at higher altitudes, which might be caused by the spatial distance between the two platforms. However, BELUGA cannot resolve values below around $\varepsilon < 10^{-6}$ m$^2$ s$^{-3}$, because the sonic on BELUGA has a higher measurement resolution limit (influenced by the sampling frequency and noise floor).

Dissipation rates for BELUGA and DH2 are further compared for all flight times when both platforms operated simultaneously (flights on 12, 15, 21, and 26 July 2020, compare Fig. 1), covering a large range of turbulence intensities. Figure 4 compares averaged dissipation rates in 10 m height bins (ascents and descents) for both platforms and the vertical and horizontal component for BELUGA. All data points as a whole show a clear, near-linear relation between DH2 and BELUGA dissipation rates. $\varepsilon_u$ from the DH2 seems to fit better with $\varepsilon_w$ from BELUGA. This might be due to the fact that BELUGA has

a lower resolution for the horizontal component and does not resolve smaller $\varepsilon$ for this component. However, when comparing





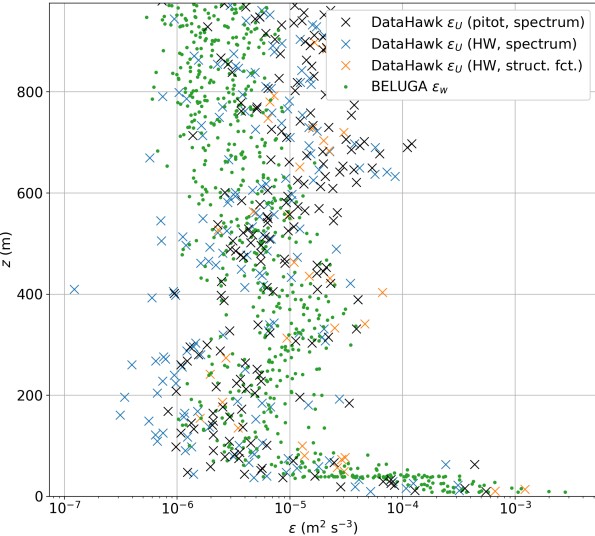

**Figure 3.** Comparison of different methods for calculating $\varepsilon$: Vertical profile (ascent) for 26 July 2020. For DH2, $\varepsilon$ is derived with the spectral method from pitot and hotwire measurements, and for comparison with the second-order structure function from hotwire measurements. $\varepsilon$ for BELUGA is derived with the second-order structure function for the vertical wind velocity component $w$.

different measurement platforms one should consider the spatial distance and time difference between the observations and potentially fast-changing ABL conditions.

### 2.3.2 Wind speed variance

The variance of a wind velocity component is another variable to describe a turbulent flow and is needed, together with the
turbulent dissipation rate, to apply Eq. (4). For the DH2, the hotwire wind velocity fluctuations would be most suitable to calculate variances in discrete time segments because the hotwire is less disturbed by motor vibrations than the pitot. However, the unstable hotwire calibration with a varying hotwire calibration coefficient would cause artificial variances. On the other hand, the distortions in the pitot airspeed data do not allow a simple variance estimation from time series segments, since the high-frequency spectral peaks create artificial variance as well.
Therefore, variances are calculated from integrated pitot-airspeed power spectra, while excluding distorted frequencies. The power spectra are calculated for 2 s intervals (an example is given in Fig. 2). A low-pass spectral filter is applied to exclude the high-frequency peaks due to motor vibrations. The cut-off frequency for the low-pass filter varies for each individual spectrum depending on the frequency range of the spectral peaks and is determined as follows: After detecting the noise floor in the spectrum, an $f^{-5/3}$ slope is fitted to the data points above the noise floor. Then the standard deviation from this curve is
calculated starting with the lowest-frequency points. If subsequent points deviate more than 10 % of the standard deviation, these points are excluded. The highest frequency of the "valid" data points is the cut-off frequency up to which the spectral

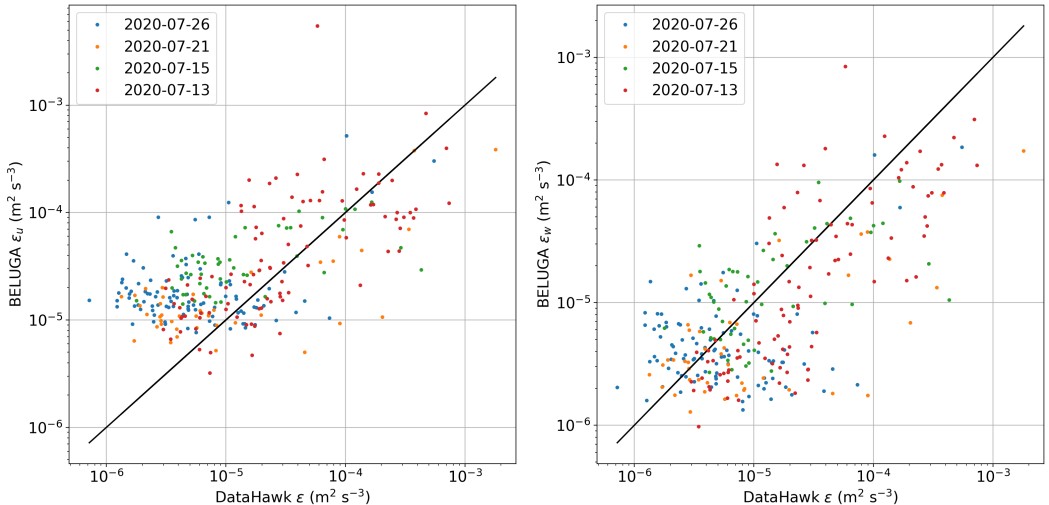

**Figure 4.** Comparison of dissipation rates $\varepsilon$ for BELUGA and DH2 for all concurrent flight times. Each data point represents a 10 m height interval with averaged $\varepsilon$ for the daily data shown in Fig. 1. For BELUGA, dissipation rates are calculated for the horizontal wind component $U$ and the vertical component $w$. The black line represents the 1:1 relation.

variance is calculated. The cut-off frequency is commonly around or above 10 Hz (in the example spectrum in Fig. 2 the cut-off frequency is 35 Hz). Each 2 s-spectrum provides one value for its variance. A larger time window than 2 s would include lower-frequency contributions, for example, 5 s segments start to show airspeed variations due to the flight pattern circles.

For BELUGA, variances are derived directly from time series segments for one wind velocity component. Turbulent fluctuations are separated from the larger-scale ABL structure by applying a high-pass 20th order Bessel filter with a filter window of typically 10-50 s length (Egerer et al., 2019). Variances are calculated in a rolling window on the vertical profile. The selected filter window determines the included scales and therefore the magnitude of resulting variances. In Fig. 5, filter windows of 5, 15 and 30 s are tested for BELUGA variances on a vertical profile for the vertical wind component $w$. The comparison shows

that the magnitude of variances is relatively insensitive to the analysis window length (the difference in variances between non-turbulent regions and the turbulent surface layer is two orders of magnitude, whereas the difference between the different filter windows is much less than one order of magnitude). Instead, the window mainly determines the vertical resolution of the variance. To compare the BELUGA method to DH2 variances, Fig. 5 adds BELUGA variances calculated from 2 s detrended time series segments (as used for DH2 variances). This is equivalent to spectral variances without applying any filter. The ver-

tical structure on the vertical profile agrees well for rolling variances in different window sizes and the 2 s detrended segments. The magnitude seems to agree well with a 15 s rolling filter window. As a result, variances for BELUGA are calculated with a 15 s high-pass filtered time series. The variance derived from DH2 measurements as described above are added in Fig. 5. The general magnitude and vertical structure compare well to BELUGA measurements, but the DH2 variances fluctuate more. This might be due to the varying cut-off frequency, single undetected spectral peaks or the fact that the DH2 generally moves more



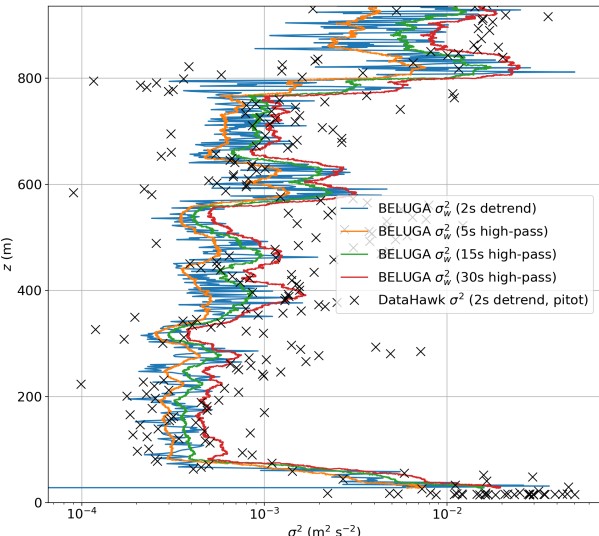

**Figure 5.** Comparison of different methods for calculating wind velocity variances $\sigma^2$: Vertical profile (descent) for 13 July 2020. The blue line represents BELUGA variances calculated from non-overlapping, detrended 2s segments. The orange, blue and red lines show BELUGA variances calculated in a rolling window after applying a high-pass filter to the time series. The black crosses result from DH2 2s-pitot filtered spectra as described in the text.

than BELUGA. The BELUGA 2 s variances also vary more than for larger time windows, which might explain a part of the DH2 variance fluctuations.

Figure 6 compares DH2 and BELUGA variances for all concurrent flights similar to Fig. 4 for dissipation rates. For BELUGA, variances are calculated for the horizontal and vertical wind components. The resolution limit (noise floor) for the horizontal component is higher, therefore values for $\sigma_u^2 < 10^{-3}$ m$^2$ s$^{-2}$ are locked to a value near the noise floor level of around $\sigma_u^2 = 10^{-3}$ m$^2$ s$^{-2}$. The influence of the measurement resolution limit is more obvious for variances than for dissipation rates. Above the BELUGA resolution limit, there is a relation between DH2 and BELUGA variances across all variance levels, despite the inevitable spatial and temporal differences in the measured parameters.

### 2.3.3 Richardson number

Stability is of central importance to describe the vertical structure of the ABL and is used in this study for parameterizations of the turbulent Prandtl number and anisotropy. Different parameters exist to describe the stability of a flow, for example, the bulk, gradient and flux Richardson number or the Monin-Obukhov length. In this study, we use the gradient Richardson number as a stability parameter. This number can describe the ABL structure locally (e.g. multiple inversions) because it does not depend on surface conditions. Further, it can be derived from vertical profile measurements of mean parameters. The gradient Richardson

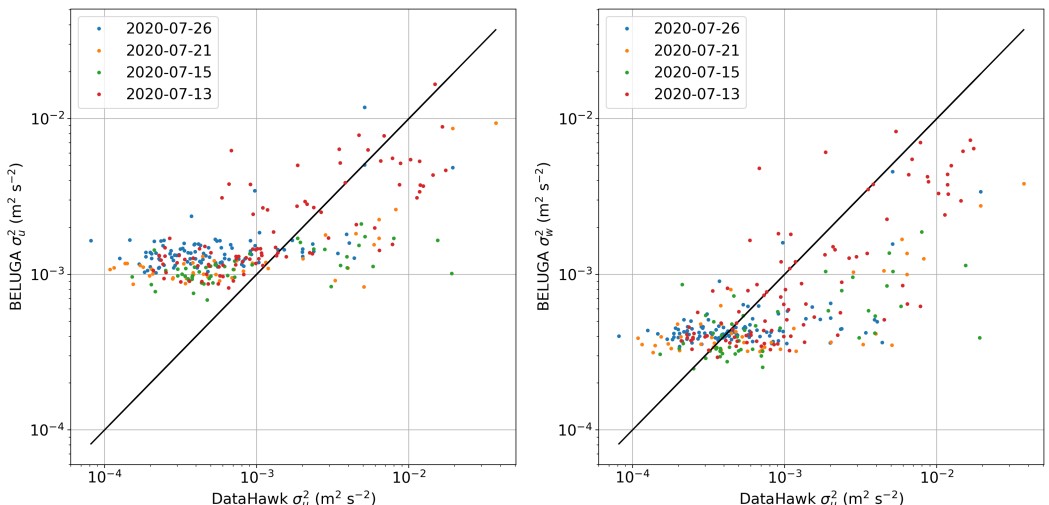

**Figure 6.** As Fig. 4, but for variances from DH2 and BELUGA.

number represents the ratio of buoyancy (with the Brunt-Väisälä frequency $N$) to wind shear $S$

$$\mathrm{Ri}_g = \frac{N^2}{S^2} = \frac{g}{\overline{\theta}_v} \cdot \frac{\partial\theta_v/\partial z}{(\partial U/\partial z)^2}. \tag{10}$$

When calculating wind and temperature gradients for $\mathrm{Ri}_g$, the measured temperature and wind profiles must be averaged in a defined time window. For BELUGA, a 20 s window is selected to provide 10 m vertical resolution at climb speeds of around $0.5\,\mathrm{m\,s}^{-1}$. DH2 ascends and descends at around $2\,\mathrm{m\,s}^{-1}$. A 10 s averaging window for temperature is selected as a compromise between vertical resolution and horizontal averaging on the flight pattern circles. Different from temperature, the DH2 wind profile is averaged over 20 s to exclude artifacts from wind changes on the helix flight pattern and from extreme bank angle changes. Outliers are excluded on the resulting $\mathrm{Ri}_g$ profile and the profile is again smoothed. Figure 7(a) shows a distribution of derived $\mathrm{Ri}_g$ for DH2 and BELUGA for one day during MOSAiC. For both platforms, the distribution of $\mathrm{Ri}_g$ peaks at values just above zero and shows a comparable density distribution. The DH2 distribution is slightly flatter than the one for BELUGA. However, the Ri-outlier problem (Sorbjan and Grachev, 2010) – the ratio of very small gradients is ambiguous and it becomes hard to differentiate stable and low-wind conditions – is especially present in conditions encountered in the Arctic. Therefore, large $\mathrm{Ri}_g$ have to be treated with caution. A critical Richardson number value $\mathrm{Ri}_c$ between 0.25 and 1 (Miles, 1961; Abarbanel et al., 1984) is assumed to differentiate conditions with high turbulence (Ri<$\mathrm{Ri}_c$) and strongly stable conditions (Ri>$\mathrm{Ri}_c$). Values for $\mathrm{Ri}_g$ in Fig. 7(a) cover both stable and unstable conditions. However, turbulence can still be present beyond $\mathrm{Ri}_c$ (Sukoriansky et al., 2006), shown by a large number of meteorological and oceanographic observations (e.g., Kondo et al., 1978; Yagüe et al., 2001; Mack and Schoeberlein, 2004). Further, Jozef et al. (2022a) found that a value of $\mathrm{Ri}_g$=0.5 to 0.75 on the vertical profile can be used to determine the ABL height based on DH2 MOSAiC data.

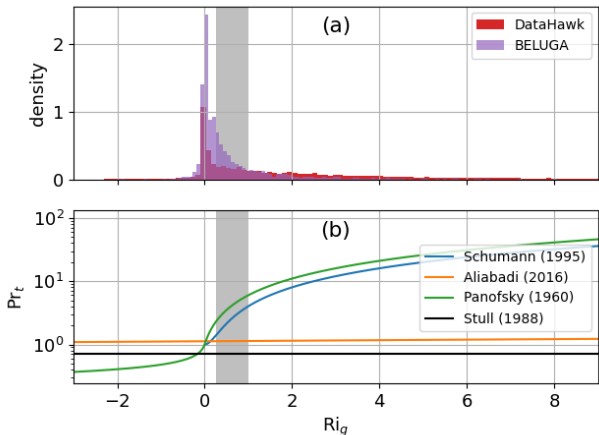

**Figure 7.** Histograms (a) of all measured gradient Richardson numbers $\mathrm{Ri}_g$ for BELUGA and DH2 for flights on 13 July and (b) different parameterizations for $\mathrm{Pr}_t = f(\mathrm{Ri}_g)$. The range for critical Richardson numbers is shown as the gray-shaded area.

### 2.3.4 Turbulent Prandtl number

The turbulent Prandtl number $\mathrm{Pr}_t$ describes the ratio of momentum transfer to heat transfer and is used in this work to apply the parameterization in Eq. (4) to the turbulent heat transport. $\mathrm{Pr}_t$ is a function of the flow itself, and more precisely of the

stability of the flow. The controversy about a quantitative description of $\mathrm{Pr}_t$ in relation to a stability parameter is still ongoing (Li, 2019; Grachev et al., 2007). Most studies agree that $\mathrm{Pr}_t \approx 1$ for turbulent flows ($\mathrm{Ri} < \mathrm{Ri}_c$). The behavior of $\mathrm{Pr}_t$ for stable flows is less clear and depends on the selected stability parameter. Many studies agree that $\mathrm{Pr}_t$ increases with increasing stability when plotting $\mathrm{Pr}_t$ versus the gradient Richardson number $\mathrm{Ri}_g$ (Kondo et al., 1978; Kim and Mahrt, 1992; Yagüe et al., 2001; Monti et al., 2002; Galperin et al., 2007). Grachev et al. (2007) found that $\mathrm{Pr}_t$ increases with increasing $\mathrm{Ri}_g$,

but decreases with increasing flux Richardson number $\mathrm{R}_f$, surface-based bulk Richardson number and the Monin-Obukhov stability parameter $z/L$ (all measures for increasing stability) although using the same data. Yagüe et al. (2001) found no clear stability dependence when using $z/L$ as a stability parameter. Opposed to other studies, Sorbjan and Grachev (2010) found that $\mathrm{Pr}_t$ decreases with $\mathrm{Ri}_g$. According to Howell and Sun (1999) and Grachev et al. (2007), $\mathrm{Pr}_t$ is even less than one when plotted against the Monin-Obukhov stability parameter.

In this study, we rely on $\mathrm{Ri}_g$ as a stability parameter even though it is prone to self-correlation and the "Ri-outlier-problem" (Grachev et al., 2007). Other parameters such as $z/L$ and the surface-based $\mathrm{Ri}_b$ assume a classical ABL and do not cover more complicated structures such as multiple inversions. Li (2019) compare different relations of $\mathrm{Pr}_t$ to $\mathrm{Ri}_g$ and conclude that a number of field and laboratory experiments and numerical simulations show an increasing, asymptotic behavior of $\mathrm{Pr}_t$ with $\mathrm{Ri}_g$ in stable conditions in the form of $\mathrm{Pr}_t/\mathrm{Pr}_{t,\mathrm{neutral}}$ (Bange and Roth, 1999; Vasil'ev et al., 2011; Aliabadi et al., 2016). The

study also compares two analytical functions for the relationship based on direct numerical simulations (Venayagamoorthy and Stretch, 2010) and large eddy simulations (LES) for laboratory experiments (Schumann and Gerz, 1995), which agree well





with experimental data from other studies. These functions are close to an earlier formulation of Panofsky et al. (1960), which is used in Hanna (1968). Figure 7b compares the different formulations to the simplified constant value of $\text{Pr}_t$=0.7 (Stull, 1988).

Because most of these studies are tied to very specific conditions often confined to the surface layer, we use the $\text{Pr}_t$ - $\text{Ri}_g$

relation of Aliabadi et al. (2016). This relation was derived in clear-air turbulence in the Arctic lower troposphere using aircraft measurements up to a 3 km altitude. Different stability regimes including counter-gradient fluxes were covered, yielding the formulation

$$\text{Pr}_t^{-1} = \frac{a}{1 + b \cdot \text{Ri}_g} \qquad (11)$$

with a=0.89 and b=0.01 (Fig. 7b). We use this parameterization because it is based on the suitable parameter $\text{Ri}_g$ and on airborne

measurements in the Arctic exceeding the surface layer, and it is a conservative estimate between other parameterizations and the simplified assumption $\text{Pr}_t \approx 0.7$.

Hence, the DH2 provides the necessary measurements of $T$, $\sigma_u{}^2$ and $\varepsilon_u$ to estimate the turbulent heat flux. It remains open that the DH2 provides the horizontal component, whereas the vertical component is needed for the method discussed above. Therefore, the next section examines anisotropy.

### 2.3.5 Anisotropy

Turbulence properties in the ABL behave differently depending on their orientation in the flow field. Isotropy is more likely to be found at smaller scales, and the scales at which a flow becomes anisotropic is influenced by stability and different other factors. Generally, anisotropy is favored by strong stability (with low turbulence) and depends on the height above the surface. Strongly turbulent flows are more isotropic than less turbulent flows; horizontal modes dominate in anisotropic flows with high

Richardson numbers (Mauritsen and Svensson, 2007). Galperin et al. (2007) showed that turbulence in an otherwise stable environment is influenced by anisotropy and internal waves. Close to the surface, horizontal mixing becomes dominant due to the spatial limitations of vertical eddies.

The $K$ parameterization in Sect. 2.2 is based on turbulence measures of the vertical wind component $w$, but the DH2 provides the horizontal wind component $U$. The slant profiles are assumed to provide horizontal measurements because of

the small effective angle to the horizontal plane of around $8°$. For using the DH2 measurements, isotropy cannot be assumed, because the stable ABL is predominantly anisotropic, with the horizontal component dominating. Therefore, we aim to describe anisotropy depending on a stability parameter so that the vertical wind turbulence estimates in Eq. (5) can be replaced by the horizontal turbulence estimates measured by the DH2. While some studies describe a qualitative relation of anisotropy and stability (Mauritsen and Svensson, 2007; Nowak et al., 2021), no quantitative parameterization of these parameters exists (to

the authors' knowledge). Therefore, we parameterize anisotropy depending on stability based on available MOSAiC data from other platforms. Anisotropy is described by a coefficient A for variances and dissipation rates:

$$A_{\sigma^2} = \frac{\sigma_w^2}{\sigma_U^2} \quad \text{and} \quad A_\varepsilon = \frac{\varepsilon_w}{\varepsilon_U}. \qquad (12)$$





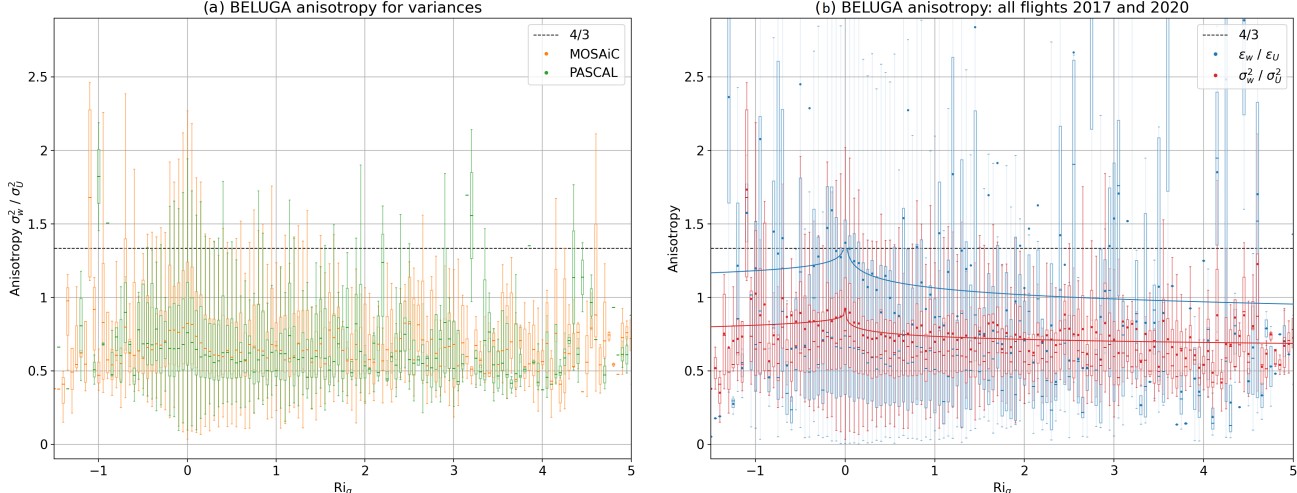

**Figure 8.** Anisotropy in relation to stability (expressed by $\mathrm{Ri}_g$) from BELUGA measurements for the MOSAiC and PASCAL campaign: (a) Comparison of anisotropy for variances between MOSAiC and PASCAL data and (b) anisotropy for variances and dissipation rates from all MOSAiC and PASCAL data. The box plots for each $\mathrm{Ri}_g$ interval include the first quartile to the third quartile of the data, with a line at the medians and a dot at the means. The mean values are used to fit a function of the form $A(\mathrm{Ri}_g) = a \cdot \pm \mathrm{Ri}_g^{(1/c)} + b$ (the function is fitted to a version of the plot with a higher-resolved $\mathrm{Ri}_g$).

While the coefficient is well suited to describe turbulence in stronger turbulence, for strong stability with weak turbulence the quotient of small values for $w$ and $U$ becomes prone to errors. We use $\mathrm{Ri}_g$ as the stability parameter. BELUGA measurements
are consulted for the sought-after parameterization because they provide estimates for variances and dissipation rates in both horizontal and vertical directions and data are collected on a vertical profile. While the tower provides the same sort of measurements, these are obviously influenced by the surface, which complicates the comparison to the DH2 measurements. All BELUGA flights during MOSAiC are considered for the anisotropy parameterization. To verify a more universal relation, data from a previous BELUGA campaign on Arctic sea ice in 2017 are included (PASCAL; Egerer et al., 2019; Wendisch et al.,
405 2019).

Figure 8 shows results for the relation of A and $\mathrm{Ri}_g$ in the form of box plots for discrete $\mathrm{Ri}_g$ bins. In Fig. 8(a) data from the two included campaigns agree well for variance anisotropy ratios $A_{\sigma^2}$ and show more anisotropy (with dominant horizontal fluctuations, meaning A<1) for stronger stability (high $\mathrm{Ri}_g$) and A closer to isotropy (A=1) for weaker stability ($\mathrm{Ri}_g$<0.25). In very unstable conditions ($\mathrm{Ri}_g$<0), A decreases again. For $A_\varepsilon$ the data are more scattered and agree less between the two
campaigns (not shown), but still show a comparable shape of the distribution. If all campaign data are plotted together, a root function for both the $A_{\sigma^2}$–$\mathrm{Ri}_g$ and $A_\varepsilon$–$\mathrm{Ri}_g$ relation can be fitted to the means of the box plots. These relations (shown in Fig. 8b) are used with the DH2 horizontal measurements to provide an estimate of the vertical component based on the





measured horizontal component and Ri$_g$. Applying these anisotropy–stability relations from BELUGA measurements allows
for using the DH2 measurements in Eq. (4) and calculating vertical flux profiles.

## 3 Vertical profile measurements for exemplary days

This section presents DH2 vertical profiles for three case studies with different ABL conditions. Two cases were obtained in
July 2020, with concurrent measurements from DH2 and BELUGA. During the first case, clear-sky ABL conditions on 26 July
were shaped by a strong persistent high-pressure system over the Barents Sea and a significant warm and moist air intrusion
from the South-East (Lonardi et al., 2022). In contrast, on 13 July a high-pressure system over the North Pole caused colder
and calm conditions with a liquid cloud layer (Lonardi et al., 2022). The third case on 9 April with a single-layer, mixed-phase
cloud is associated with an anomalously cold period associated with air masses coming from the North (Rinke et al., 2021).

### 3.1 26 July 2020: Stably stratified ABL with weak turbulence

Many of the DH2 measurement days during MOSAiC are characterized by stable stratification of the ABL. The 26 July case
is one example of these conditions and is selected for analysis because of concurrent measurements of DH2 and BELUGA up
to 1000 m. General conditions on this day were clear-sky with some intermittent low-level fog or haze near the surface evident
in radar data. Lidar data occasionally show very thin high clouds formed in aerosol-rich layers probably without significant
impact on the lower atmospheric structure. Both DH2 and BELUGA measurements up to 1000 m (Fig. 9) show a similar ABL
structure with a surface-based temperature and humidity inversion between the surface and 200 m. Above the inversion, the
ABL is slightly stable throughout the profile with nearly-constant $q$. The air mass is warm and moist with potential temperature
up to 20 °C and $q$ between 5–6 g kg$^{-1}$. Wind speed is as well fairly constant throughout the profile, not exceeding 5 m s$^{-1}$
below 800 m. Meteorological measurements from both platforms agree well with the radiosonde and tower measurements.

Dissipation rates and wind speed variances indicate surface-induced turbulence within the inversion layer. Above the inver-
sion turbulence gets very weak with values of $\varepsilon < 10^{-5}$ m$^2$ s$^{-3}$. At this point, the BELUGA measurements already fall below
the sonic instrument's noise level, which becomes evident by the barely-varying values throughout the profile. DH2 shows
more $\varepsilon$ and $\sigma^2$ variations within several layers of tens of meters thickness. The panels for $\varepsilon$ and $\sigma^2$ also compare the DH2 "hor-
izontal" component (actual slant profile measurements) and "vertical" component (measurements corrected with the anisotropy
relation in Sect. 2.3.5). The difference between those components (corresponding to the anisotropy factor in Fig. 8) is much
less than the variation between the turbulent surface layer and the stable layer above (equal to two orders of magnitude); the
general profile is not altered by using either of the components. Anisotropy close to the surface also reflects in the tower mea-
surements: these show higher variances (by almost one order of magnitude) for the horizontal component than for the vertical
component because of larger-scale fluctuations included in the 13 min averaging interval. DH2 and BELUGA do not cover
these larger-scale fluctuations due to their measurement principle (the averaging interval is restricted on the vertical profile).
For the mast dissipation rates, no difference between the horizontal and vertical component is evident because dissipation rates
are derived in the inertial subrange where low-frequency contributions are not included. The turbulent exchange coefficient $K$





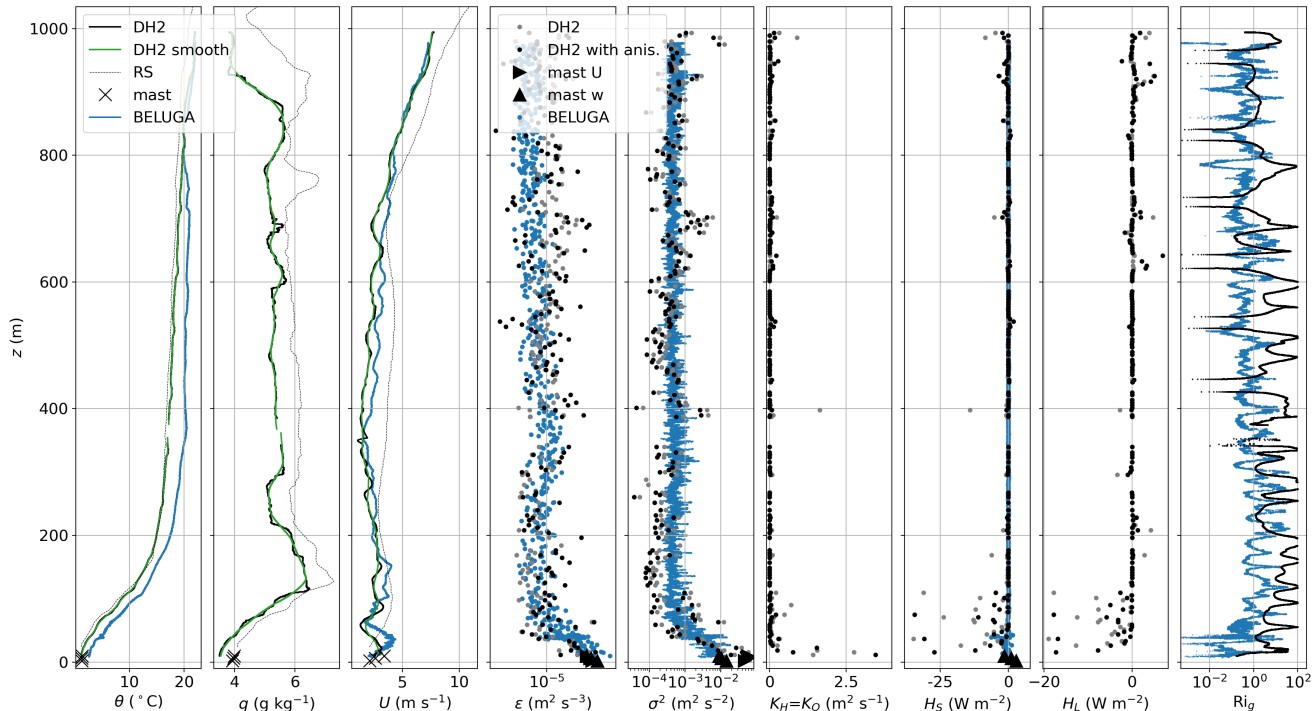

**Figure 9.** Vertical profiles for 26 July (first ascent) for DH2 (black lines and dots) and BELUGA (blue lines and dots) with radiosonde (thin dotted lines) and tower measurements (crosses and triangles near the surface) as reference. The panels show potential temperature $\theta$, specific humidity $q$, horizontal wind speed $U$, dissipation rate $\varepsilon$, wind speed variance $\sigma^2$, turbulent exchange coefficients $K_H$ and $K_Q$, turbulent heat fluxes of sensible heat $H_S$ and latent heat $H_L$, and gradient Richardson number $\mathrm{Ri}_g$. The green lines show the smoothed DH2 profiles for gradient calculations. The grey dots for $\varepsilon$, $\sigma^2$, $K$ and flux values are derived from horizontal DH2 measurements without anisotropy correction.

is close to zero throughout the vertical profile (due to the low turbulent motions), but increases near the surface. This applies as well to sensible and latent heat fluxes: these are close to zero throughout the profile and turn negative inside the inversion. Here, enhanced turbulent motions mix heat and moisture downward along the mean temperature and humidity gradients. The larger flux values are associated with more scatter. The BELUGA sensible heat flux profile looks similar, but with smaller flux values in the inversion. This is probably caused by the limited averaging time of 10 s for the covariances. However, the mast

flux magnitudes with the 13 min averaging time indicate a negative flux with similar magnitude at 11 m height and slightly positive fluxes below. Gradient Richardson numbers are below $\mathrm{Ri}_c$ close to the surface, matching the turbulence profiles. In the stable region above, $\mathrm{Ri}_g$ indicates several thin layers of increased turbulence. These might be natural or a result of the quotients of shallow temperature and wind velocity gradients. To conclude, the measurements of all platforms draw a consistent picture of the ABL conditions for stable stratification with increased turbulence near the surface. The turbulent flux profiles resulting

from measured turbulence parameters and mean gradients complement the picture reasonably.



## 3.2   13 July 2020: Decoupled, cloud-driven mixed layer

The 13 July 2020 case is characterized by a decoupled mixed layer with a stratocumulus cloud in its upper part. Radar measurements show a persistent cloud layer just above 1 km height, slightly varying in height and thickness (Shupe et al., 2022).
The DH2 recorded two profiles up to the cloud base; BELUGA flew a profile almost simultaneously up to above the cloud
top. The DH2 and BELUGA profile measurements in Fig. 10 show the strong 7 K temperature inversion capping the cloud
layer between 1100 m and 1300 m. Additionally, weaker temperature inversions are observed within the lowest 70 m near the
surface and barely visible at 600 m and 800 m height. For this day, no reliable humidity data are available from the DH2, but
the radiosonde profile suggests a relatively wet ABL with $q = 3.5 - 4$ g kg$^{-1}$ and multiple weak inversions. Wind speed is
around 6 m s$^{-1}$ inside and below the cloud, decreasing to a minimum at 570 m, and with a shallow 6 m s$^{-1}$ low-level jet at the
top of the surface-based temperature inversion, evident in DH2 and BELUGA measurements.

Dissipation rates and wind speed variances again show a similar turbulent structure and define the cloud-driven mixed
layer with increased turbulence inside and below the cloud between 800 m to 1300 m. Observations of vertical velocity
and spectral width from the cloud radar (not shown) also support this general structure. This mixing is caused by cloud-top
radiative cooling, which drives buoyant, upside-down shallow convection extending below cloud base. Due to relatively weak
turbulence, the cloud-driven mixed layer does not extend below about 800 m, such that this layer is decoupled from lower
atmospheric layers and the surface below. Below the mixed layer, turbulence is very weak and thus, this stable layer decouples
the mixed layer from the surface. At around 600 m, another thin layer of increased turbulence (more pronounced in DH2
data than for BELUGA) seems to be associated with an intermittent and thin secondary cloud layer occasionally visible at
different levels below the primary cloud in the radar data (Fig. 1). At the bottom, the surface-driven turbulent layer extends
up to about 75 m within the surface-based temperature inversion. The profile of turbulent exchange coefficients $K$ shows
increased values where turbulence is highest: close to the surface, within the cloud-driven mixed layer and at the secondary
cloud layer near 600 m, leading to negative (downward) sensible heat fluxes in these layers. DH2 flux estimates fluctuate much
more than BELUGA flux estimates, probably due to the longer averaging times for BELUGA. The negative sensible heat
fluxes at around 800 m are basically a detrainment of heat from the cloud-driven mixed layer. The mixed layer is relatively
warm ($\theta \approx 10°C$) and probably had little interaction with the melting sea-ice surface (at $\sim 0°C$) over the course of its advective
path. This is similarly described in Shupe et al. (2013): A relatively warm, moist air mass moves over the sea ice and remains
decoupled from the surface partly because of the vast difference in the thermodynamic state of the cloudy mixed layer versus
the surface layer. Some of the warmth of the layer is lost due to radiative cooling at the cloud top, and some is lost by downward
mixing, which effectively increases the energy content of the layer between 0–800 m and also contributes to sensible heating
of the surface, as seen in the surface layer. The magnitude of DH2 near-surface fluxes agrees well with tower-derived fluxes,
despite the disparity in averaging intervals. However, even the relatively reliable tower estimates differ by 5 W m$^{-2}$ between
the individual measurement heights, which suggests strong variability in the surface layer. Altogether, the observations of
ABL conditions with a decoupled, cloud-driven mixed layer are in agreement with previous observations (Shupe et al., 2013;
Sotiropoulou et al., 2014) and add information about downward turbulent fluxes at layer interfaces.



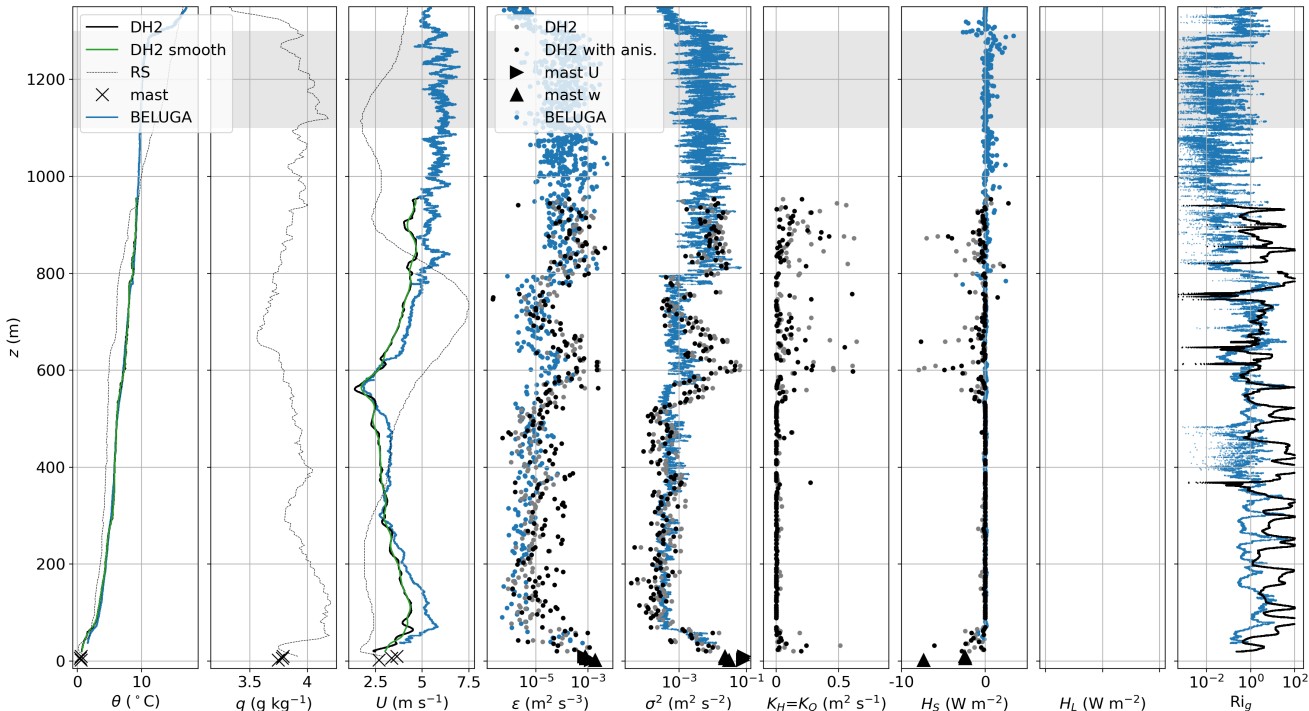

**Figure 10.** Vertical profiles for 13 July (first descent). Panels are as in Fig. 9, but no reliable $q$ and $H_L$ data from DH2 are available for this case.

### 3.3  9 April 2020: Cloud and wind-shear driven turbulence

9 April 2020 is a case with a mixed-phase cloud typical for the Arctic ABL. At the time of the DH2 flight, radar and lidar data show a persistent liquid cloud with cloud base at about 900 m and ice crystal precipitation (and sublimation) below (Fig. 1). The radar Doppler spectral width shows turbulent mixing as a result of cloud radiative cooling and buoyancy extending below the cloud base down to approximately 500 m (not shown). The DH2 flew a vertical profile up to just below cloud base at 900 m, but no BELUGA flights are available. Figure 11 depicts the rather complicated ABL structure recorded by the DH2. The thin (100-m thick), near-neutral surface layer is capped by stable stratification above with several smaller temperature inversions between the surface and 600 m, some weak overturning at 500 m and slightly stable to near-neutral conditions above 600 m. Throughout, temperatures are very low: down to -22 °C at the surface and -14 °C near the cloud base. The specific humidity profile resembles the temperature profile; the liquid water content is very low due to the cold temperatures. The profile reveals an apparent moisture inversion above the surface to 300 m, highlighting the important role of advective moisture (and the limited surface source of moisture at this time of the year). The wind profile exhibits a weak low-level jet in the stably stratified region between the surface and 600 m with a maximum wind speed of 7.5 m s$^{-1}$ at 400 m.

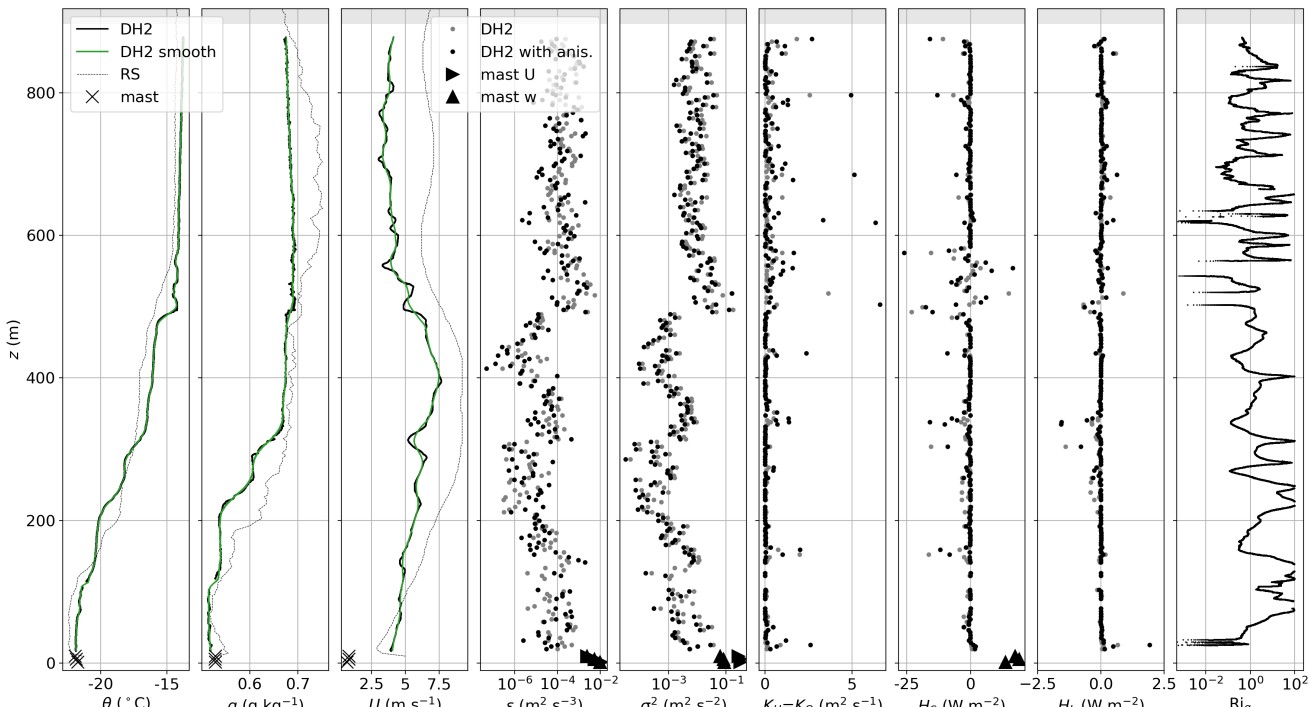

**Figure 11.** Vertical profiles for 9 April (descent). Panels are as in Fig. 9.

The turbulence profiles for $\varepsilon$ and $\sigma_U^2$ feature several turbulence maxima probably generated from three different sources: (i) surface-based turbulence – the vertical profiles seem to clearly continue the mast measurements, (ii) cloud-driven turbulence evident as a constant turbulence magnitude between cloud base and the lower boundary of the near-neutral layer at 500 m, and (iii) shear-induced turbulence by the LLJ with a local minimum at the jet core and increased values below and above at 300 m and 500 m. An increase in turbulent dissipation on the upper and lower edges of the LLJ has been observed in previous studies relating LLJs and turbulence (e.g., Banta et al., 2006; Smedman et al., 1993). Throughout the profile, the turbulence is strongest at the interface of the cloud mixed layer with the upper bound of the LLJ at 500 m, where the temperature profile also shows the overturning. At this altitude, the turbulent heat fluxes are most pronounced as well with a negative (downward) sensible and latent heat flux at the top of the temperature and humidity inversion. The variability in $H_S$ at the bottom of the cloud-driven mixed layer reflects the slight variation in $\theta$ between 500-600 m. Probably, the base of the mixed layer is not static but varies in space and time, leading to some inconsistencies and turbulent exchanges. The presence of some upward sensible heat fluxes above suggests the interaction of multiple layers in that zone (also subtly seen in the temperature profile). Comparing with the radiosonde suggests that there is an evolution in this layer just above 500 m. Downward-oriented heat fluxes also occur just below the jet core and a stronger upward-directed flux (15 W m$^{-2}$) is observed in the near-neutral surface layer. Lastly, the Ri$_g$ profile again shows very small values in the high-turbulence regions.



For all three cases presented here, the observations represent typical ABL structures in the Arctic that have been observed previously. The DH2 observations add valuable information about the turbulence vertical structure and turbulent fluxes in
regions with pronounced turbulence. Although flux magnitudes seem to be consistent with surface flux measurements, the absolute values of fluxes should be treated with caution, because the variance estimates only include small scales due to the short time records. Nonetheless, the method presented provides a robust idea of the vertical profile shape of turbulent fluxes.

## 4   Discussion and conclusions

This work and the case studies herein demonstrate the potential of the DH2 measurements to analyze turbulence and turbulent
fluxes in the Arctic ABL as observed during MOSAiC. The flux gradient method with the parameterization of the turbulent exchange coefficient is an established method to derive vertical profiles of turbulent fluxes. The method of Hanna (1968) has been applied with sUAS measurements before by Platis et al. (2016) for studying particle fluxes by means of $K_m$. The present study extends the Hanna (1968) method to $K_H$ and $K_Q$ for sensible and latent heat fluxes by parameterizing the relation of different turbulent exchange coefficients via $\mathrm{Pr}_t$. For this, we apply a parameterization of $\mathrm{Pr}_t$ depending on stability derived
from airborne measurements in the Arctic ABL (Aliabadi et al., 2016). If the flux method will be applied to DH2 measurements in other locations, the selection of the $\mathrm{Pr}_t$-parameterization might have to be re-evaluated depending on prevailing stability conditions. Another novelty presented here is the anisotropy parameterization in the vertical ABL profile depending on the gradient Richardson number as a stability parameter. With this parameterization, the fluctations of vertical wind components result from measurements of the horizontal component. The anisotropy parameterization was derived from airborne sonic
anemometer measurements during MOSAiC and another Arctic field campaign. As a result, the extended flux method allows for estimating vertical profiles of turbulent fluxes based on measurements of the one-dimensional, high-resolution horizontal wind speed together with a low-resolution temperature/ humidity measurement. Hence, measurement instrumentation can be kept relatively simple, which is advantageous with the limited payload and battery capacity of sUAS.

However, the applied method is subject to several limitations. Generally, the flux gradient method is rather suitable for stable
stratification (which mostly applies to conditions in this study) than for unstable conditions, and counter-gradient fluxes are not represented by the method. However, studying stable ABLs brings different challenges: With shallow vertical gradients and small flux magnitudes, small perturbations in the measured parameters increase relative flux errors. Also, the length and time scales included in the flux estimates are restricted by the averaging time for variances (Eq. 4 and the anisotropy parameterization) since the flux magnitude is proportional to the square of variances (when assuming correctly estimated dissipation
rates). The estimates for variances include only time scales smaller than the averaging time, so the derived fluxes represent the small-scale turbulent transport. Further, short averaging intervals, compared to integral time scales, increase random and systematic errors of variances and fluxes (Lenschow et al., 1994). Nonetheless, DH2 flux magnitudes near the surface agree well with eddy covariance fluxes from a co-located, ground-based tower. Moreover, the ABL is assumed to be stationary over the course of the flight. Other limitations result from the measurement principle of sUAS: First, the DH2 helix flight pattern
produces a slant profile instead of a true vertical profile. We assume the slant profile measurements as horizontal and average





these over a certain height interval. If the interval is too small, horizontal heterogeneity might appear as 'vertical' fluctuations (Balsley et al., 2018). On the other hand, the interval cannot be too long or the method will not achieve the desired vertical resolution. Second, most DH2 flights are located outside of clouds and the flights are limited to lower-wind conditions, which excludes some case analyses that might be especially interesting when studying the Arctic ABL. Further errors might occur due

to DH2-specific issues and the measurement conditions: The wind estimation is inaccurate under certain conditions of extreme flight dynamics (de Boer et al., 2022; Doddi et al., 2022) and the wake of the ship during MOSAiC might have influenced the measurements near the surface. Lastly, the anisotropy parameterization relies on relatively few BELUGA measurements in stable stratification. For future applications, the authors recommend extending and verifying this parameterization.

Despite all these limitations, the DH2 results agree well with established measurement methods like meteorological flux
towers and radar. This provides confidence in the obtained results and offers novel insights into turbulent transport processes in the Arctic ABL: (i) The case studies in this work represent typical Arctic ABL structures observed in previous studies. Nonetheless, high-resolution vertical profile measurements are rare, and the DH2 may offer very detailed insights into turbulent exchange processes. (ii) These vertical profile details also provide important context for the evolution of the surface energy budget, which then controls the sea ice thermodynamic state and melt. In particular, this is evident in the 13 July case where
downward sensible heat fluxes warm the near-surface layer and support ice melt. (iii) In the past, these vertical transfers of turbulent heat fluxes within the ABL typically have been inferred from model simulations as very few measurements were available of this type. The results from the presented method will be essential for evaluating LES studies examining the energy and moisture budgets associated with clouds and cloud-driven mixed layers (e.g. Solomon et al., 2014; Neggers et al., 2019).

The methods shown in this study will be extended to further cases of interest, which requires careful examination of the
available measurements for each individual case. Turbulent flux profiles from the DH2 are available for the whole operation period during MOSAiC from winter to the melt season. The resulting vertical profiles of turbulent fluxes can be analyzed concerning different ABL and sea ice conditions, including the influence of atmospheric stability, stratification, clouds, leads, and melt ponds to understand the complex interactions between ABL processes, the surface energy budget, and sea ice. Some of these cases can support LES studies, where these new observation-based perspectives will add unique new constraints on
cloud, turbulent, and moisture processes. All of these insights will help to advance our understanding of how turbulent fluxes influence the interactions between the Arctic atmosphere and surface.

*Data availability.* DataHawk2 meteorological data: Arctic Data Center (Jozef et al., 2021, 2022b). BELUGA turbulence data: PANGAEA (Egerer et al., 2021), Met tower data: Arctic Data Center (Cox et al., 2021). Cloud Radar data: DOE ARM Archive (Johnson et al., 2021).

*Author contributions.* UE performed the turbulence data analysis with support from DL and AD. GJ, GdB, JJC, RC and JH conducted the
DataHawk2 flights and post-processed and published the meteorological data. MDS was the ATMOS team lead during MOSAiC and com-





plemented the case descriptions with additional ground-based data. HS was the BELUGA PI during MOSAiC. UE compiled the manuscript with contributions from all co-authors.

*Competing interests.* The authors declare that they have no competing interests.

*Acknowledgements.* This work was supported by the CIRES Visiting Fellows Program that is funded by the NOAA Cooperative Agreement

with CIRES, NA17OAR4320101. Data used in this manuscript were produced, as part of the international Multidisciplinary drifting Observatory for the Study of Arctic Climate (MOSAiC) with the tag MOSAiC20192020. We thank all persons involved in the expedition of the Research Vessel *Polarstern* during MOSAiC in 2019–2020 (AWI_PS122_00) as listed in Nixdorf et al. (2021). M.D.S. was supported by the National Science Foundation (OPP-1734551), DOE Atmospheric System Research Program (DE-SC00021341), and NOAA Physical Sciences Laboratory (NA22OAR4320151). G.d.B. was supported by the US National Science Foundation Office of Polar Programs

(OPP 1805569), the US Department of Energy Atmospheric Systems Research program (DE-SC0013306) and the NOAA Physical Sciences Laboratory. Part of the work was supported by the Deutsche Forschungsgemeinschaft (German Research Foundation)—project number 268020496—TRR 172, within the Transregional Collaborative Research Center "ArctiC Amplification: Climate Relevant Atmospheric and SurfaCe Processes, and Feedback Mechanisms (AC)3" in sub-project A02. A subset of data was obtained from the Atmospheric Radiation Measurement (ARM) User Facility, a U.S. Department of Energy (DOE) Office of Science User Facility Managed by the Biological and

Environmental Research Program. The authors would like to thank Andrey Grachev and Chris Fairall for fruitful discussions.



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
