# Peer review of "Estimating turbulent energy flux vertical profiles from uncrewed aircraft system measurements: Exemplary results for the MOSAiC campaign"

_Atmospheric Measurement Techniques, 2022_

## Referee Comment (RC2)

Referee Report on
**Estimating tubulent energy flux vertical profiles fro uncrewed aircraft system measurements: Exemplary results for the MOSAiC campaign**

The authors present an overview of their approach to derive turbulent statistics, particularly dissipation rate, $\varepsilon$, normal Reynolds stresses (variances of velocity fluctuation), eddy diffusivities, and heat and moisture fluxes using uncrewed aerial system (UAS) measurements. I found the paper to be well written and found that it does a good job of reviewing most of their calculation approaches and provides a careful and honest examination of the authors' results. This information is potentially useful for other researchers interested in extracting similar statistics or for researchers who are interested in the details of the calculations when examining data produced by the aircraft. I therefore feel that the article is suitable for publication in Atmospheric Measurement Techniques.

One major concern I had was in the description and usage of the turbulent kinetic dissipation rate, $\varepsilon$. Although by definition this is a scalar quantity (describing the rate of viscous dissipation of the scalar quantity turbulent kinetic energy), the authors repeatedly refer to it as having different components ($\varepsilon_U$ and $\varepsilon_w$) as well as having anisotropy between these components. I believe that this may stem from a misunderstanding of equation 8 in the manuscript, whereby they describe the wavenumber as a scalar property, whereas it is actually a vector quantity. Given that they are applying Taylor's frozen flow hypothesis, it appears that Equation 8 is actually describing the longitudinal wavenumber spectrum, in which case $E(k)$ describes the energy content of the velocity component parallel to the wavenumber, $k$. This then means that the Kolmogorov constant $\alpha$ for the $w$ component of velocity should actually be closer to 0.65 (from Pope (2000), page 232) leading to a difference in $\varepsilon$ estimates of about 30%, when using the frequency spectrum to estimate it, which could explain the difference the authors are seeing in their values of $\varepsilon_w$ an $\varepsilon_u$. Alternativley the different noise floor for the $U$ and $w$ components of velocity from the sonic on the BELUGA could also impact their estimates from the second order structure function, if not properly accounted for. Either way, I found the description of $\varepsilon$ as having different components to be confusing.

I am also concerned with their parameterization of anisotropy, as anisotropy is a relic of the boundary conditions at which the turbulence is produced it is unlikely to be readily parameterized, particularly near a surface. Furthermore, limiting to relatively small scales the calculation of $\sigma_u^2$ and $\sigma_v^2$ could also be impacting the value of their anisotropy estimates due to the difference in the expected inertial subrange scaling for longitudinal vs lateral wavenumber spectra mentioned above.

Other, minor, comments that I also feel should be addressed before publication are as follows:

1. [Line 100] It would be valuable for the authors to provide the radius of their spiral ascent/descent, as this impacts the validity of how they apply Taylor's hypothesis in Eq. 8. (Tighter spirals are not likely to provide effective approximations of longitudinal spectra).

2. [Line 117] How is the 2 s used to estimate the spectrum determined? For a 15 m s$^{-1}$ airspeed, this would only correspond to eddies of wavelengths shorter than 30 m. One would expect integral scales to roughly increase with altitude, such that eddies on the order of 100 m or larger could be present for the boundary layers shown in Figs. 9,10 11 and the full energy content not captured in these quantities.

3. [Line 118] I am curious about the frequency response for the pitot probe. Typically the pitot tube would experience some attenuation at high frequency due to viscous damping and resonance at certain frequencies due to the transducer cavity. In Hamilton et al (2022) only the hot-wire probe is cited as having the O(1kHz) frequency response so the use of the pitot probe for these calculations requires some justification.

4. [Line 143] "and temporary 23 m made " should perhaps be "and temporarily 23 m were made".

5. [Line 150] There should be a space between 23 and m.

6. [Line 178] $\sigma_w$ should be specifically defined to refer to the standard deviation of the vertical component of velocity. Similarly, the quantity $\varepsilon_w$ should also be more specifically defined since $\varepsilon$ is later defined as the dissipation rate.

7. [Equation 9] What is the quantity $\overline{U}_\tau$ and how does it relate to $\overline{U}$ used in equation 8?

8. [Line 219] How does the local dissipation rate $\varepsilon_\tau$ differ from $\varepsilon$?

9. [Line 245] Referring to Fig.2, it would appear that the frequencies above 40 Hz are elevated above the -5/3 slope. How is this impacting the dissipation rate calculation, given that it is calculated using the full 2-400 Hz spectrum?

10. [Line 249] Given that Doddi (2021) is a thesis, perhaps it would be best to have some of the details of the spectral analysis included in this paper?

11. [Line 295] As I understand the procedure being described, the authors are anchoring the fit to the lowest frequency point in the spectrum and using that as an initial reference for discarding points whose least-squares value is too high. However, is this justified given that the lowest wavenumbers are likely to be the least accurate (i.e. it represents the amplitude of only a single wave)?

12. [Figure 5] Assuming a wind speed of 5 m s$^{-1}$ and UAS airspeed of 15 m s$^{-1}$, it might be worthwhile noting that the comparison here shows energy content in scales on the order of 30 m for the UAS, and between 10 and 150 m for the BELUGA. I would argue that this explains the trends observed between the different high-pass filter window lengths, but the best comparison to the UAS would be expected from the the 5 s window (depending on horizontal wind speed).

13. [Line 351] It is perhaps not surprising that that behavior of the turbulent Prandtl number is less clear for stable flows, given that its definition presumes the existence of turbulence (implicit in the eddy viscosity/eddy diffusivity definition).

14. [Line 378] "It remains open that the DH2 provides the horizontal component," horizontal component of which quantity? This sentence is unclear.

15. [Line 384] The statement that strongly turbulent flows are more isotropic than less turbulent flows is only appropriate for high wavenumbers. At small wavenumbers, strongly turbulent flows can be very anisotropic (e.g. in the neutral turbulent boundary layer).

16. [Line 406] The authors refer to A, but previously had defined only $A_\sigma$ and $A_\varepsilon$.

17. [Line 429] $q$ was not previously defined.

18. [Figure 9] Which value of $\sigma^2$ is being plotted, previously had defined $\sigma_U^2$ and $\sigma_w^2$. In the text (e.g. Line 503) it is referred to as $\sigma_U^2$

19. [Line 443] What is meant by this statement? Equations 8 and 9 are only valid in the inertial subrange.

20. [Line 506] The acronym LLJ was not defined.

---

## Author Comment (AC1)

**Author's response to review #1 of research article AMT-2022-314**

We would like to thank anonymous referee #1 for their constructive review, which significantly improved the quality of the manuscript.

**Answers to specific reviewer comments**

**Major comments**

1. **Figure 2 shows a 2 s spectrum for pitot and hotwire fluctuations. Is 2 seconds interval spectrum sufficient for such a turbulence structure analysis? For example, a typical ECOR system uses at least 10 mins data.**

   We are aware that with this short averaging time, we exclude larger scales of the energy spectrum and only capture contributions from smaller eddies when calculating variances. The averaging time is always a compromise that is impacted by various practical reasons. In this paper, we were originally using consistent 2s interval lengths for HW calibration, $\varepsilon$ calculation, and $\sigma^2$ calculation. The consistent interval ensures that we can use $\varepsilon$ and $\sigma^2$ in the same equation (Eq. 8 in the manuscript) with a common resolution.

   Further reasons for using a relatively short interval are: 1) Shorter intervals provide the highest spatial resolution. The main purpose of our vertical profile measurements is to resolve thin layers. When using longer averaging times, the vertical profiles are blurred; 2) At larger intervals, airspeed variations due to the DH spirals start to be visible; 3) For HW calibration, the calibration coefficient varies a lot and the averaging interval should be as short as possible; 4) For $\varepsilon$, the interval is of minor importance because its value can be estimated from any spectral data in the inertial subrange; 5) For $\sigma^2$, the averaging interval has the greatest impact, but as the resulting variance is found to be rather insensitive to the interval (see Fig. 5), the short interval of the other parameters is used for $\sigma^2$ as well (for the DH2).

   The main difference to an ECOR system is that we are not aiming to capture low-frequencies in the spectrum, but instead, we aim to resolve a vertical profile and the turbulent exchange between shallow layers in the ABL. The plot copied below shows a time series of estimated variances for different DH2 averaging intervals, showing an expected decrease in resolution with increasing averaging time, yet the magnitude of the variance increases only slightly relative to the range in variance over time (altitude). After careful consideration of all influencing factors, we decided to increase the averaging time to 5s as the best compromise for this work. The impacted parameters throughout the manuscript are changed correspondingly.

[Figure]

We included this discussion in the manuscript in Sect. 2.3.2, line 306ff.

2. **Line 200: How does the author select the value of C? What is the appropriate range of the C values? Similar situation for the appropriate range for the Kolmogorov constant (in line 222). Will you please provide a guideline for those parameters' determination? For example, are those constants unique to the Arctic environment or general cloudy conditions?**

Thank you for this comment. The constant C has to be determined from observations and depends on stability. The value of C=0.35 of earlier studies was confirmed by the observations in Hanna 1968, which encompass a large variety of (stability) conditions. We revisited the literature and decided to use the value of C=0.41 instead, which seems to be more suitable for stable conditions (Lee 1996). We've revised the calculations and updated the text passage (line 206ff).

The Kolmogorov constant is universal and does not depend on the Arctic environment or cloudy conditions. Experimental data support that for the three-dimensional energy spectrum $\alpha=1.5$. For one-dimensional measurements, the corresponding constant depends on whether the component measured is longitudinal or lateral to the flow (relative wind): $\alpha_{long}=0.5$, $\alpha_{lat} = 4/3*\alpha_{long} = 24/55*\alpha \approx 0.65$ (Pope 2000, p.232). As we use the spectral method for the DataHawk2 horizontal measurements, we use $\alpha_{long}=0.5$ in this study.

3. **How many flights do you compare the DH2 measurements with BELUGA? Do you have some statistics to confirm the DH2 performance?**

The flights illustrated in Figure 1 of the manuscript were used to compare DH2 measurements and BELUGA (in total four days each with one or two profiles per platform). Please also see the comment below for a comparison between the two platforms for these flights individually. These were the only flights during which the DH2 and the BELUGA sonic anemometer flew concurrently. Therefore, we do not have more robust statistics from MOSAiC to compare DH2 and BELUGA measurements.

4. **Figures 9, 10, and 11 show that the temperature difference between DH2 and BELUGA (for the potential temperature profiles) is 2-4 C. That is relatively large. Do you have any explanations for the data quality and the meaningfulness of using the comparison? Do you have other ground comparisons to determine the temperature measurement's uncertainty range? Similar concerns with the dissipation rate, wind speed variance and the gradient Richardson number. They were plotted on a log scale, and it is hard to understand how accurately the new approach derived parameters compared with BELUGA.**

The temperature difference in Fig. 9 is in the range of 2-4 $^{O}$C only below 800m altitude. Above, the temperature profiles agree very well. Therefore, we assume that this difference is not a systematical measurement offset, but is caused by spatial and temporal heterogeneity of the ABL. Also, Fig. 10 shows an almost perfect agreement between DH2 and BELUGA temperature measurements. The plots also include a comparison to meteorological mast measurements, which as well do not show a systematic offset. Figure 11 does not have BELUGA measurements. A detailed comparison of dissipation rates and variances derived from BELUGA and DH2 is shown in Figs. 4 through 6, pointing out that the DH2 can resolve smaller turbulent eddies than BELUGA, but within the resolved scales, the measurements agree.

Further, DH2 measurements of temperature, wind, and humidity were compared to those from the radiosondes as an established platform by Jozef et al. (2022), and they found that DH2 and radiosonde profiles of the aforementioned variables were similar to each other, such that features including ABL height, low-level jets, and inversions were in agreement between DH2 and radiosonde measurements taken at approximately the same time. For example, when comparing ABL height from DH2 and radiosonde observations within ~3 hours of each other, no significant difference at the 5% significance level was found. Additionally, Hamilton et al. (2022) provide detailed statistics on the performance of the DH2 when compared to radiosonde observations within 1 hour of the DH2 launch during MOSAiC, showing reasonable agreement of temperature and wind. The Jozef et al. (2022) paper provides in the supplementary figures the profiles of bulk Richardson number from all DH2 flights and the corresponding radiosonde (closest radiosonde to DH2 launch, within ~3hrs). These plots show that $Ri_b$ from DH2 and radiosonde are generally in good agreement. The $Ri_b$ profiles from the radiosondes were less noisy than that from the DH2, but this can be attributed to smoother profiles due to the lower vertical resolution of measurements from the radiosonde versus the DH2.

We have added comparisons to radiosoundings to the revised manuscript in Sect. 2.1.2, line 111ff.

**Minor comments**

- **Equation 9 used equation 10 in Siebert et al. (2006) for the u component. How do you derive C2 =2.6 for vertical velocity components?**

  For the longitudinal spectrum: $C_{2,long}$=2 (as also noted in Siebert 2006). For the lateral spectrum: $C_{2,lat}$ = 2* 4/3 ≈ 2.66 (Pope 2000). In isotropic conditions, the Kolmogorov theory predicts a 4/3 ratio between the spectra of lateral and longitudinal wind velocity components in the inertial subrange (Kaimal et al. 1972).

- **In section 2.3.1, the structure of this section is confusing. Before Line 232, the author introduced the method used by Siebert et al. (2006), then starting in line 332, "a different established method is applied to derive dissipation rates." Please list the equations for the other method. What are the connections between the two methods? Do you plan to compare them? Or do they complement each other? Which method is more suitable for the Arctic environment? What are the pros and cons of choosing each method in Fig 3?**

  The formulation was misleading, we do not use a different method other than the ones introduced: for DH2 the spectral method, and for BELUGA the second-order structure function. We have clarified this in the revised text in section 2.3.1, line 273ff. We also added in line 224: "Both techniques estimate dissipation rates at inertial subrange scales and are independent on the larger scales." Fig.3 compares the different methods applied to the DH hotwire data for one day where the hotwire data quality allows applying both methods.

---

## Author Comment (AC2)

**Author's response to review #2 of research article AMT-2022-314**

The comments of referee #2 were very helpful in advancing and revealing some issues in the manuscript We very much appreciate the time and effort the referee dedicated to reviewing the manuscript.

**Answers to specific reviewer comments**

**Major comments:**

- **One major concern I had was in the description and usage of the turbulent kinetic dissipation rate, ε. Although by definition this is a scalar quantity (describing the rate of viscous dissipation of the scalar quantity turbulent kinetic energy), the authors repeatedly refer to it as having different components ($\varepsilon_U$ and $\varepsilon_w$) as well as having anisotropy between these components. I believe that this may stem from a misunderstanding of equation 8 in the manuscript, whereby they describe the wavenumber as a scalar property, whereas it is actually a vector quantity. Given that they are applying Taylor's frozen flow hypothesis, it appears that Equation 8 is actually describing the longitudinal wavenumber spectrum, in which case E(k) describes the energy content of the velocity component parallel to the wavenumber, k. This then means that the Kolmogorov constant α for the w component of velocity should actually be closer to 0.65 (from Pope (2000), page 232) leading to a difference in ε estimates of about 30%, when using the frequency spectrum to estimate it, which could explain the difference the authors are seeing in their values of $\varepsilon_w$ an $\varepsilon_u$. Alternativley the different noise floor for the U and w components of velocity from the sonic on the BELUGA could also impact their estimates from the second order structure function, if not properly accounted for. Either way, I found the description of ε as having different components to be confusing.**

  We agree that our original description of anisotropy for ε was misleading in the sense that anisotropy does not exist for ε itself, but rather for the one-dimensional spectral formulation $E(k) = \alpha \cdot \varepsilon^{2/3} \cdot k^{-5/3}$. We use the spectral method for the DataHawk ε estimation from fluctuations in the relative wind, hence this is a longitudinal measurement with α =0.5. For BELUGA, we use both the $w$ and u components (separately) to estimate ε from the structure functions, where the instrument's $u$ is aligned (weathervanes) into the relative wind. We changed some formulations in the manuscript, replacing "components of ε" with ε estimated from the vertical ($w$) /horizontal ($u$) spectrum or structure-function.
  Further, we slightly modified our approach and hope that it is more straightforward now:
  - o   For DH2, ε is calculated with the longitudinal spectrum
  - o   For BELUGA, ε is estimated from lateral and longitudinal second-order structure functions with different C for longitudinal and lateral components (we originally had used the same longitudinal C for both). We call the resulting dissipation rates $\varepsilon_U$ and $\varepsilon_w$ based on the wind component they originate from.
  To evaluate the anisotropy of the flow, we relate $\varepsilon_w$ to $\varepsilon_U$ (depending on stability expressed by Ri) and apply this ratio to ε from DH2 because the Hanna 1968 method is based on the vertical velocity spectrum. The mean ratio $\varepsilon_w/\varepsilon_U$ at Ri=0 (Fig. 8b) is now closer to 1 as would be expected for isotropic conditions, rather than  4/3 as it was before. For BELUGA, the noise floor is accounted for when calculating ε by accepting only exponents in a certain range when fitting Eq. (9).

- **I am also concerned with their parameterization of anisotropy, as anisotropy is a relic of the boundary conditions at which the turbulence is produced it is unlikely to be readily parameterized, particularly near a surface. Furthermore, limiting to relatively small scales the calculation of $\sigma^2_u$ and $\sigma^2_v$ could also be impacting the value of their anisotropy estimates due to the difference in the expected inertial subrange scaling for longitudinal vs lateral wavenumber spectra mentioned above.**

We agree that anisotropy cannot be entirely parameterized. We added some text about this in the manuscript in Sect. 2.3.5, line 412ff: "… anisotropy depends on many influence factors beyond stability, especially near the surface, and cannot be entirely parameterized. However, finding an empirical correlation between anisotropy and layer stability provides a useful way to predict when anisotropy could be expected." Please see also the comment above about how we aim to use this formulation. Since we derive the anisotropy ratios from BELUGA by including the same scales as for DH2, we do not describe anisotropy of the entire spectrum, but only for the scales of interest for our application.

**Minor comments:**

1. **[Line 100] It would be valuable for the authors to provide the radius of their spiral ascent/descent, as this impacts the validity of how they apply Taylor's hypothesis in Eq. 8. (Tighter spirals are not likely to provide effective approximations of longitudinal spectra).**

   Diameters of the spiral ascents/descents are 150m to 200m, we've now added this information (line 103).

2. **[Line 117] How is the 2 s used to estimate the spectrum determined? For a 15 m s$^{-1}$ airspeed, this would only correspond to eddies of wavelengths shorter than 30 m. One would expect integral scales to roughly increase with altitude, such that eddies on the order of 100 m or larger could be present for the boundary layers shown in Figs. 9,10 11 and the full energy content not captured in these quantities.**

   That is correct, the original 2s intervals do not capture the full energy content. The main reason we are using short intervals is that we aim to resolve the vertical structure with shallow layers, as they occur typically in the Arctic. We increased the interval to 5s as a compromise. Please also see our response to major comment no. 1 of reviewer # 1.

3. **[Line 118] I am curious about the frequency response for the pitot probe. Typically the pitot tube would experience some attenuation at high frequency due to viscous damping and resonance at certain frequencies due to the transducer cavity. In Hamilton et al (2022) only the hot-wire probe is cited as having the O(1kHz) frequency response so the use of the pitot probe for these calculations requires some justification.**

   The pressure sensor manufacturer does not specify the sensor bandwidth, and as the reviewer notes, the pitot tube and associated tubing to the sensor can result in filtering of the airspeed signal. However, we use a very small pitot tube, and very short tubing to the sensor (< 5 cm) to avoid these issues. The best evidence of this is seen in the power spectra of airspeed fluctuations, which do not show any inherent roll-off in addition to the expected f$^{-5/3}$ inertial range cascade up to the Nyquist frequency (400Hz for 800Hz sampling). In this typical example spectrum (below) of the pitot probe,

the high frequencies are influenced by the noise floor and motor vibrations above about 40 Hz, and the roll-off at 300 Hz is due to anti-alias filtering of the pitot signal. These effects do not influence the ε estimates due to the use of spectral averages at lower frequencies ("calibration points"). We added some of this discussion in Sect. 2.1.2, line 127ff.

[Figure]

4. **[Line 143] "and temporary 23 m made " should perhaps be "and temporarily 23 m were made".**

   Thanks, done.

5. **[Line 150] There should be a space between 23 and m.**

   Done.

6. **[Line 178] $\sigma_w$ should be specifically defined to refer to the standard deviation of the vertical component of velocity. Similarly, the quantity $\varepsilon_w$ should also be more specifically defined since ε is later defined as the dissipation rate.**

   We changed $\varepsilon_w$ to ε in the formulation and defined $\sigma_w$.

7. **[Equation 9] What is the quantity $U_\tau$ and how does it relate to U used in equation 8?**

   Please see the comment below.

8. **[Line 219] How does the local dissipation rate $\varepsilon_\tau$ differ from ε?**

   The index τ served only to show that local dissipation rates are estimated in defined time periods τ. As this is the case for other parameters and the spectral method as well, we agree that using the index is confusing and we removed it.

9. **[Line 245] Referring to Fig.2, it would appear that the frequencies above 40 Hz are elevated above the -5/3 slope. How is this impacting the dissipation rate calculation, given that it is calculated using the full 2-400 Hz spectrum?**

   Frequencies that are impacted by peaks in the spectra, caused by motor vibrations, are excluded from the spectral estimations. The 2-400Hz range is the maximum range in which the dissipation rate is calculated, but the frequencies impacted by the peaks and the noise floor are excluded. We formulated that more clearly in Sect. 2.3.1, line 252f.

10. **[Line 249] Given that Doddi (2021) is a thesis, perhaps it would be best to have some of the details of the spectral analysis included in this paper?**

The details of the spectral analysis are included in the text passage above the reference to the Doddi (2021) thesis. We have clarified this in the revised text.

11.  **[Line 295] As I understand the procedure being described, the authors are anchoring the fit to the lowest frequency point in the spectrum and using that as an initial reference for discarding points whose least-squares value is too high. However, is this justified given that the lowest wavenumbers are likely to be the least accurate (i.e. it represents the amplitude of only a single wave)?**

Two things to note here. 1. The lowest frequencies contain the fewest data points (highest frequencies - most data points). The spectra are frequency binned (equal bin width in log space), this equally weights all the frequency bins. The fitting is conducted on the binned spectra. 2. A more thorough analysis (not included in this study) of the effect of windowing on each frequency bin suggests a very small effect on the lowest frequencies which negligibly affects the fit.

12. **[Figure 5] Assuming a wind speed of 5 m s$^{-1}$ and UAS airspeed of 15 m s$^{-1}$, it might be worthwhile noting that the comparison here shows energy content in scales on the order of 30 m for the UAS, and between 10 and 150 m for the BELUGA. I would argue that this explains the trends observed between the different high-pass filter window lengths, but the best comparison to the UAS would be expected from the the 5 s window (depending on horizontal wind speed).**

BELUGA and DH2 use different averaging times for variances (DH2 5s intervals, BELUGA 15s intervals, based on established methods from prior work and practical reasons), because of which the lengths scales included in the variance calculations are similarly in the order of 75m. We included in Sect. 2.3.2 in the paragraph about BELUGA variances and comparison to DH2 (line 325ff): "Note that since the airspeed of the DH2 is about 15m/s and the typical wind speed measured by BELUGA is about 5m/s, using a 5s analysis window for DH2 measurements would be equivalent to a 15s analysis window for BELUGA measurements in terms of the wind field scales included in the variance estimates." However, the included effective length scales depend on the mean wind speed for BELUGA.

13. **[Line 351] It is perhaps not surprising that that behavior of the turbulent Prandtl number is less clear for stable flows, given that its definition presumes the existence of turbulence (implicit in the eddy viscosity/eddy diffusivity definition).**

We added this remark.

14. **[Line 378] "It remains open that the DH2 provides the horizontal component," horizontal component of which quantity? This sentence is unclear.**

We hope that the revised sentence  is more clear: "It remains open that the DH2 airspeed measurements represent the near-horizontal vector component of velocity fluctuations (due to the small slant-path angle), whereas the vertical component is needed for the method discussed above."

15. **[Line 384] The statement that strongly turbulent flows are more isotropic than less turbulent flows is only appropriate for high wavenumbers. At small wavenumbers, strongly turbulent flows can be very anisotropic (e.g. in the neutral turbulent boundary layer).**

We re-formulated the passage, now it reads: "Generally, anisotropy is favored by strong stability (with low turbulence); horizontal modes dominate in anisotropic flows with high Richardson numbers (Mauritsen and Svensson, 2007). Galperin et al. (2007) showed that turbulence in an otherwise stable environment is influenced by anisotropy and internal waves. Anisotropy also depends on the height above the surface: close to the surface, horizontal mixing becomes dominant due to the spatial limitations of vertical eddies."

16. **[Line 406] The authors refer to A, but previously had defined only $A_\sigma$ and $A_\varepsilon$.**

    Changed.

17. **[Line 429] q was not previously defined.**

    $q$ is defined above Eq. 3.

18. **[Figure 9] Which value of $\sigma^2$ is being plotted, previously had defined $\sigma^2_U$ and $\sigma^2_w$. In the text e.g. Line 503) it is referred to as $\sigma^2_U$**

    The new legend entries should explain which component is depicted.

19. **[Line 443] What is meant by this statement? Equations 8 and 9 are only valid in the inertial subrange.**

    Thank you, hopefully this is a better formulation: "… because $\varepsilon$ is a measure of energy dissipation at small scales in the inertial subrange which are more isotropic."

20. **[Line 506] The acronym LLJ was not defined.**

    Definition added.